# DNA origami presenting the receptor binding domain of SARS-CoV-2 elicit robust protective immune response

Esra Oktay[1], Farhang Alem[2], Keziah Hernandez[2], Michael Girgis[1], Christopher Green[3], Divita Mathur[4], Igor L. Medintz[3], Aarthi Narayanan ⓘ [2✉] & Remi Veneziano ⓘ [1✉]

Effective and safe vaccines are invaluable tools in the arsenal to fight infectious diseases. The rapid spreading of severe acute respiratory syndrome coronavirus 2 (SARS-CoV-2) responsible for the coronavirus disease 2019 pandemic has highlighted the need to develop methods for rapid and efficient vaccine development. DNA origami nanoparticles (DNA-NPs) presenting multiple antigens in prescribed nanoscale patterns have recently emerged as a safe, efficient, and easily scalable alternative for rational design of vaccines. Here, we are leveraging the unique properties of these DNA-NPs and demonstrate that precisely patterning ten copies of a reconstituted trimer of the receptor binding domain (RBD) of SARS-CoV-2 along with CpG adjuvants on the DNA-NPs is able to elicit a robust protective immunity against SARS-CoV-2 in a mouse model. Our results demonstrate the potential of our DNA-NP-based approach for developing safe and effective nanovaccines against infectious diseases with prolonged antibody response and effective protection in the context of a viral challenge.

[1] Department of Bioengineering, George Mason University, Fairfax, VA 22030, USA. [2] National Center for Biodefense and Infectious Diseases, George Mason University, Manassas, VA 20110, USA. [3] Center for Bio/Molecular Science and Engineering Code 6900, U.S. Naval Research Laboratory, Washington, DC, USA. [4] Department of Chemistry, Case Western Reserve University, Cleveland, OH, USA. ✉email: anaraya1@gmu.edu; rvenezia@gmu.edu

With the emergence of severe acute respiratory syndrome coronavirus-2 (SARS-CoV-2) and thus coronavirus disease 2019 (COVID-19), our healthcare systems have been facing unprecedented challenges[1]. The COVID-19 pandemic has already resulted in the death of millions around the world and forced authorities to impose mandatory quarantines to limit viral spreading[2]. SARS-CoV-2 is an emerging and highly infectious RNA virus that belongs to the β-coronavirus genus, which contains other viruses that are responsible for previous major outbreaks[3]. One key strategy to limit the spreading of SARS-CoV-2 is the development of safe and effective vaccines that can provide long-lasting immunity and protect against all variants[4,5]. Current vaccine strategies primarily use all or part of the viral spike protein (S) as an immunogen that can be delivered in various forms (e.g., mRNA, DNA plasmid, and protein)[6]. Among all of the candidate vaccines that reached clinical trials, only four have been issued an Emergency Use Authorization (EUA) or a full FDA authorization. The two mRNA-based vaccines, namely the mRNA-1273 from Moderna (Spikevax) and the BNT162b2 from Pfizer/BioNTech (COMIRNATY®) with a > 90% efficacy reported, have been granted a full FDA authorization. The viral-vector-based vaccine [Ad26.CoV2.S from Johnson & Johnson (Janssen)] with 66% efficacy and the protein nanoparticle-based vaccine Novavax-COVID-19 from Novavax have been granted an EUA[7–10]. Numerous traditional vaccine strategies that rely on either killed-inactivated or live-attenuated viruses are currently under development, some of which are in Phase III of clinical trials, but strikingly none have yet received FDA approval[11].

Other methods such as subunit vaccines are currently being developed, as they represent a safe way to deliver antigens. However, monovalent antigens are known to often trigger low immunogenicity that results in limited protection, which can be mitigated by displaying them in the multivalent form on nanoparticle (NP) carriers[12,13]. Because NP-based vaccines do not use viral genetic materials and are assembled with biocompatible materials[14], they also reduce safety concerns[15]. Additionally, their chemical nature, surface compositions, and modifications can improve vaccine stability, thereby helping to prolong their shelf-life, extend their bioavailability, and facilitate their cellular uptake[15,16]. To this day, more than 60 NP-based SARS-CoV-2 vaccines have been developed, showing the high interest for these strategies[17]. Among these candidates, most of them deliver the receptor binding domain (RBD) of the spike S1 protein or of the full spike protein[18–20]. The RBD appears to be a potent immunogen and one of the main targets of neutralizing antibodies found in vaccinated and infected people[21,22]. Interestingly, few mRNA vaccine studies have also highlighted the importance of using RBD (particularly in a trimeric form)[23] to induce strong immune response in comparison with its monomeric form or the S1 spike protein[20]. This domain is also crucial during viral infection due to its role in interacting with host cells to promote viral entry. The RBD is recognized by the angiotensin-converting enzyme 2 (ACE2) receptor and therefore plays a vital role in viral tropism and spreading to cells expressing ACE2 receptors[24]. The structural feature of the RBD has been identified as having either closed or open states[25]. In the closed conformation, three RBDs of the trimeric S protein can mask themselves to avoid recognition by the immune system. The S protein undergoes conformational changes and only one RBD reveals its binding motif to interact with ACE2 in open conformation, which is then followed by the sequential opening of the other two RBD motifs and thereby forms the trimeric S protein-ACE2 receptor complex[25–27]. Many studies have shown that a multivalent display of RBDs induces a stronger immune response and a higher level of neutralizing antibody titer in comparison with monomeric RBD[28–31].

However, the limited control offered by NPs does not allow for assessing the role of structural parameters related to antigen presentation (e.g., density, stoichiometry, nanoscale organization, and NP geometries) on cellular uptake and immunogenicity that could enable rational design of vaccines[32,33].

Scaffolded DNA origami nanoparticles (DNA-NPs), on the other hand, provide an ideal biocompatible platform for assessing antigen presentation parameters[34,35]. Indeed, DNA-NPs can be designed in any geometry and size[35,36] while permitting the patterning of biomolecules with nanoscale precision, which allows for the modulation of antigen stoichiometry and facilitates multiplexing[37,38]. Recently, DNA-NPs have been used to present antigens in controlled stoichiometry and nanoscale organization to modulate the activation of immune cells. Specifically, Veneziano et al. showed that presenting HIV-glycoprotein (eOD-GT8) antigens on DNA-NPs in various nanoscale organization and stoichiometry led to the modulation of B-cell activation by inducing the clustering of B cell receptors in vitro[37]. Thus, using DNA origami could inform the rational design of vaccines and represent a promising alternative nanocarrier for effectively and safely delivering viral antigens. In this study, we developed a DNA-NP vaccine platform (DNA-NP nanovaccine) that can accommodate multiple copies of a single immunogen and adjuvant simultaneously. We also assessed its immunogenicity and efficacy in vivo. Specifically, we repurposed a DNA pentagonal bipyramid (PB) that have been previously used to present HIV proteins antigens to study B cell activation[37].

The advantages of using this specific nanoarchitecture for presenting antigens and for subsequent cellular uptake has been discussed in previous studies[39–41]. Particularly, the almost flat surface of the PB makes it similar to oblate ellipsoidal nanoparticles that are known to be preferentially uptaken by immune cells due to the larger surface area of interaction and the receptor diffusion kinetics, which facilitate membrane wrapping and internalization based on simulations and in vitro experiments[39,40,42]. Moreover, our PB DNA origami NP also provides two surfaces with the same geometries that allow multiplexed presentation of various biomolecules simultaneously (Supplementary Fig. 1). In addition, recent studies have shown that DNA origami and particularly the pentagonal bipyramid have negligible immunogenicity, especially in regards to the stimulation of the TLR9 pathway, which is important to determine the specificity of the vaccine response induced by the antigens[43,44]. Therefore, we used the PB structure to present the RBD antigen in a trimeric form, along with multiple CpG ODN 1018 (cytosine-phosphorothioate-guanine oligodeoxynucleotides) adjuvants (Fig. 1), to assess the efficiency of the DNA-NP nanovaccine in mounting a strong and protective immune response even after two months following the second vaccine dose.

## Results and discussion

**Designing and constructing PB DNA-NPs**. Using the DAE-DALUS (DNA Origami Sequence Design Algorithm for User-defined Structures) software[36], we designed a DNA PB (Supplementary Fig. 1) with a diameter of 36.5 nm (52 base pairs [bps] edge length). The sub-100 nm size of our NP was chosen to facilitate lymph drainage and promote uptake by antigen-presenting cells, as previously demonstrated in the literature regarding the optimal size for efficiently draining to lymph nodes and current NP-based approaches such as protein and lipid NPs[45–47]. The PB was assembled with a long single-stranded DNA (ssDNA) scaffold that was produced by asymmetric PCR[48,49] (Supplementary Tables 1 and 2 and Supplementary Fig. 2) and folded with 44 staple strands (sequences of scaffold and staple

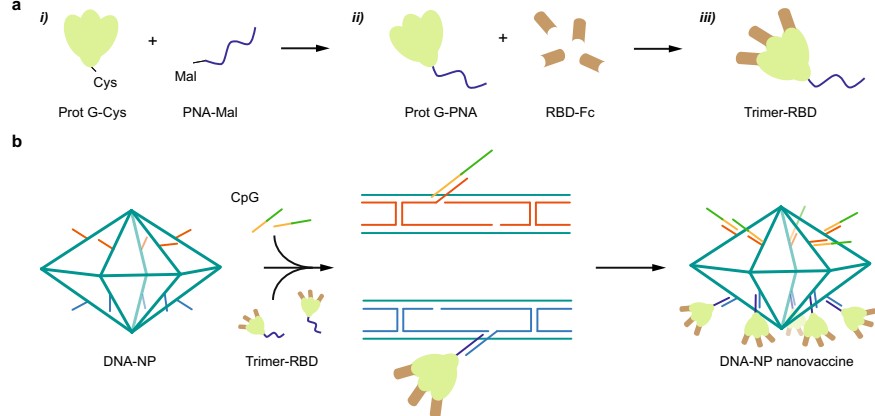

**Fig. 1 Assembly of the DNA-NP nanovaccine. a** Formation of the peptide nucleic acid (PNA)-RBD trimer by coupling three RBD-Fc on the three Fc-binding domains of a protein G (PG): **i)** A PNA strand is conjugated to a PG via maleimide chemistry (Mal). **ii)** PG-PNA is used to couple three RBD-Fc. **iii)** The trimer is purified from the free RBD-Fc. **b** Two separate faces of the PB are modified with ssDNA overhangs on defined locations to facilitate attachment of the PG-RBD complex and CpG adjuvants.

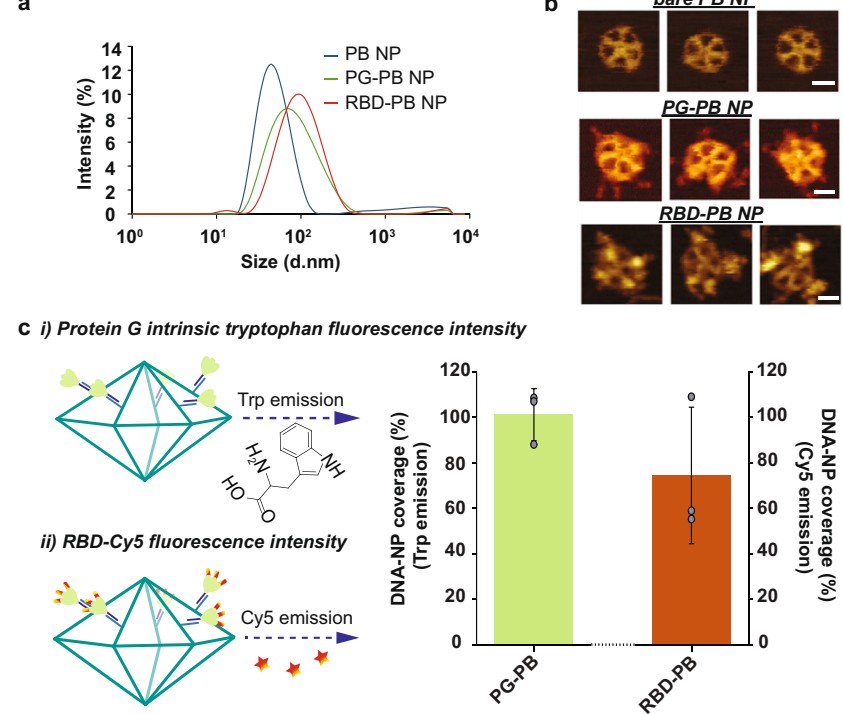

**Fig. 2 Physicochemical characterization of DNA origami nanoparticles. a** Hydrodynamic diameter measurement of DNA-NPs with DLS. **b** Atomic force microscopy imaging of DNA-NPs formation and conjugation with 3-mer RBDs (scale bar: 20 nm). **c** Fluorescence-intensity based determination of PG and RBD stoichiometry on DNA origami NP. **i)** Tryptophan fluorescence emission was used to determine the total number of PG loaded on the surface of the DNA-NPs. **ii)** Second, the stoichiometry of RBD on NP was quantified via measuring the emission of Cy5 dyes conjugated to the RBD antigens. The bar graph represents the total coverage percentage for the PG (green bar) and the 3-mer RBD (orange bar) on the DNA-NP surface normalized to the number of conjugation sites available. Data are shown as the mean ± SD ($n = 3$ independent experiments).

strands are listed in Supplementary Table 1 and Supplementary Tables 3 to 5, respectively). The PB edges were designed to display up to 10 ssDNA overhangs per face (2 overhangs per edges) at the 3'-end of selected staple strands (Supplementary Fig. 3) to anchor antigens and CpG adjuvants modified with complementary overhangs via direct hybridization. In addition, these nanoparticles offer 10 more ssDNA overhangs on the side edges for higher stoichiometry if required. These overhangs were designed to be orthogonal to each other in order to facilitate asymmetric modification of the NPs, thus enabling simultaneous presentation

of the antigens on one face and delivery of the conjugated adjuvants on the other face. The correct folding of the PB constructs was validated with agarose gel electrophoresis, atomic force microscopy (AFM), and dynamic light scattering (DLS), which showed a well-folded monodisperse particle population with a folding yield estimated at 96% (Fig. 2a, b and Supplementary Fig. 4). The diameter of the PB DNA-NP was measured with DLS at 51.2 nm for a theoretical diameter of 46.4 nm for particles including overhangs (Supplementary Fig. 4). This result is consistent with previous results reported with this type of NPs[36,37].

**Table 1 Hydrodynamic diameter of the DNA-NP with each protein attachment.**

|  | Bare PB NP (mean ± SD) | PG – PB NP (mean ± SD) | RBD – PB NP (mean ± SD) |
|---|---|---|---|
| Z-average (nm) | 47.6 ± 0.64 | 70.7 ± 0.61 | 83.6 ± 1.52 |
| PDI | 0.241 ± 0.005 | 0.261 ± 0.007 | 0.261 ± 0.005 |

*PDI* Polydispersity Index.

**Reconstitution of trimeric-RBD via Protein G-Fc conjugation strategy**. Next, we prepared the reconstituted RBD-trimer (3-mer RBD) that we used as an immunogen to display on the DNA-NPs. We used a commercially available modified version of the protein G (cys-PG) that contains a single cysteine residue on its N-terminal. A peptide nucleic acid strand (PNA) was conjugated to the cys-PG via a maleimide (Mal) group (Fig. 1a). PNA is a nucleic acid analog composed of peptide bonds and nucleobases capable of hybridizing DNA via Watson Crick base-pairing with a higher affinity than DNA:DNA hybridization[50]. The PG-PNA conjugation efficiency was validated with sodium dodecyl sulphate polyacrylamide gel electrophoresis (SDS-PAGE) (Supplementary Fig. 5) and the product purified with centrifugal filtration. The concentration of PG was estimated based on the concentration of PNA. We then reacted various ratios of RBD-Fc and PG-PNA to ensure complete reconstitution of the RBD trimer and validated their formation with native- and SDS-PAGE (Supplementary Fig. 6).

**PB DNA-NPs-presenting CpGs and RBD-trimers construction**. To control the organization and stoichiometry of the 3-mer RBDs on the surface of the PB DNA-NPs, we hybridized 10 copies of PG-PNA to specific overhangs displayed on the surface of the DNA-NPs (Fig. 1b). The 3-mer RBDs were then assembled on the antigen presenting face of the PB DNA-NPs through PG-PNA, named RBD-PB NP (stoichiometry discussed below), and 10 CpG strands were also hybridized on the opposite face of the PB DNA-NPs via hybridization on a second set of overhangs, named RBD-PB-CpG (Fig. 1b). We used the CpG ODN 1018 (cytosine-phosphorothioate-guanine oligodeoxynucleotides), which contains a fully phophorothioated backbone and has been already proven to be efficient in an FDA-approved Hepatitis B vaccine[51]. Specifically, CpG 1018 is known to stimulate Th1-biased CD4+ T cells characterized with pro-inflammatory cytokine IFN-γ secretion as a part of antiviral immunity[52,53]. After validating attachment of PG-PNA to specific overhangs via PAGE (Supplementary Fig. 7), we confirmed the attachment of the protein complex RBD-PG-PNA, which is defined as 3-mer RBD on overhangs of PB NP, along with CpG hybridization, via agarose gel electrophoresis (Supplementary Fig. 8)

To further characterize the conjugation efficiency of proteins onto the surface of the PB DNA-NP we performed DLS measurements. We expected the diameter of the DNA-NPs to increase if binding occurs on their surface. We measured and compared the diameter differences between DNA-NPs with and without PG-PNA or 3-mer RBD. The average diameter ($n = 3$ technical replicates) of NPs before and after each conjugation step is indicated in Table 1. We noted an increase in the diameter of NPs of 23.1 nm following PG-PNA hybridization and an increase of 12.9 nm following conjugation of the three RBDs on the DNA-NP through PG-PNA (Fig. 2a). This increases in size associated with a low polydispersity index (PDI) ranging between 0.241 and 0.261 (Table 1) confirmed by a sharp DLS peak validate the conjugation on the surface of the DNA-NPs and the presence of a monodisperse population of DNA-NPs (Supplementary Fig. 9).

To qualitatively assess the conjugation of the different proteins on the surface of the DNA-NPs we further used AFM imaging and compared the bare PB DNA-NP with the PG-PB NP and RBD-PB NP (Fig. 2b and Supplementary Fig. 10). The AFM images of both the PG-PB NP and the RBD-PB NP confirm the presence of proteins attached to the arms of the DNA-NP on one side only, which confirms the efficacy of our conjugation protocol. Altogether, the gel electrophoresis results, the DLS experiments and the AFM images validate the efficiency of our protocol to assemble PB DNA-NPs that can present 3-mer RBDs via PG.

**Determination of the stoichiometry of PG and RBD on DNA-NP**. To quantitatively determine the level of conjugation, we used fluorescence measurements. We first used tryptophan fluorescence assays to determine the level of PG modification on DNA-NP displaying 10 overhangs (Supplementary Fig. 11). Based on the fluorescence measurements, the percentage of coverage of the DNA-NP with PG was found to be close to 100% (i.e., 101.2 ± 11.4%, mean ± SD of $n = 3$ independent samples) (Fig. 2c). Further, by using a fluorescently labelled version of the RBD protein (cy5 labelled), we were able to measure the percentage of the coverage of the DNA-NP presenting 10 copies of PG. Our results show that about 74.4% (Fig. 2c) of the available sites (30 sites) on the PG were occupied by the RBD proteins (i.e., ~23 RBD domains for 10 overhangs) (Supplementary Fig. 12).

**Measuring the RBD:PG ratio on DNA-NP via mass spectrometry**. To further characterize our PB DNA-NPs and determine the absolute ratio between PG and RBD, we developed methods for multiple reaction monitoring (MRM) utilizing a tandem mass spectrometry (MS/MS) technique. For PG, the peptide used for quantification has a sequence of GETTTEAVDAATAEK and the targeted precursor mass-to-charge ratio (m/z) was determined at $m/z = 747.3519^{++}$ (for 2 copies of the peptide per protein). The peptide was quantified based on the transition $m/z = 1004.48^+$. For the RBD protein, the peptide used for quantification was VGGNYNYLYR and the m/z ratio for the targeted precursor was $609.7987^{++}$ (for 1 copy of the peptide per protein). The peptide was quantified based on the transition $m/z = 891.45^+$. The peak area from each transition (Supplementary Fig. 13 and Supplementary Table 6) was correlated to the standard curve and the peptide concentration was calculated. Our results indicate that the molar ratio between RBD and PG was determined to be 2.82:1 (RBD:PG) for a theoretical ratio of 3:1 (Supplementary Fig. 14), which confirmed the efficacy of our conjugation strategy.

**Quantification of the coverage of CpG on DNA-NP**. The quantification of CpGs on DNA-NPs was determined using fluorescence measurements with a fluorescein-labelled version of the CpG ODN. Using a standard curve made with the free fluorescein-CpG ODN, we estimated the CpG ODN hybridization yield to be about 80% (Supplementary Fig. 15).

**Characterization of the stability of PB DNA-NPs**. Prior to assessing the efficacy of our DNA-NP nanovaccines in vivo, we evaluated their stability in simulated physiological conditions. We used a Förster resonance energy transfer (FRET)-based assay with two FRET reporter pairs (Supplementary Table 7) located on two different edges of the PB DNA-NP. The sites selected do not interfere with the antigen or the CpG binding sites. We used fluorescein (FAM) and tetramethylrhodamine (TAMRA) as donor and acceptor dyes, respectively. The FRET pair were designed with a distance of 8 bases (~3 nm) between dyes to maximize the FRET efficiency. The stability of NPs was evaluated

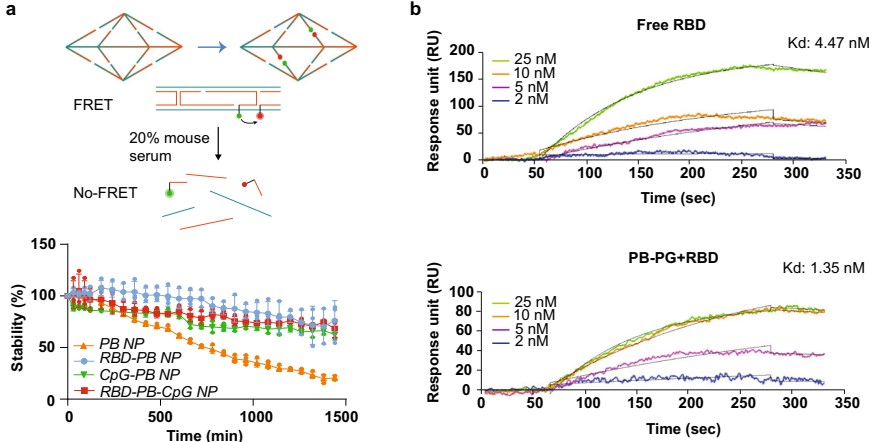

**Fig. 3 Characterization of PB DNA-NP nanovaccines. a** Determination of nanovaccines stability via an in vitro FRET-based assay. Nanovaccine samples were incubated for 24 hours in 20% mouse serum and the changes in the FRET efficiency were examined to determine stability. PB-CpG, RBD-PB, and RBD-PB-CpG NPs showed elevated stability over the entire 24-hour period in comparison to bare PB NP. The fluorescence intensity of bare PB decreased approximately 79% after 24 hours. (Orange line: bare PB NP in serum as positive control, $n = 3$ independent samples; blue line: RBD-PB nanovaccines in serum, $n = 3$ independent samples; green line: PB-CpG nanovaccines in serum, $n = 3$ independent samples; red line RBD-PB-CpG nanovaccines in serum, $n = 3$ independent samples). Data were shown as mean ± SD on graph. Two-way ANOVA was applied for analyzing mean differences and Tukey's post-hoc test was used for the stability comparisons in time between each group ($p < 0.001$). **b** Representative binding kinetics measurement via surface plasmon resonance. Four different concentrations (25 nM, 10 nM, 5 nM, and 2 nM) of free RBD-Fc monomer and trimeric RBD-PB were tested for their binding kinetics flowing over immobilized ACE2 receptor. Measured Kd between free RBD monomer and ACE2 was determined at 4.47 nM, and Kd between 3-mer RBD-PB and ACE2 receptor was determined at 1.35 nM.

| Table 2 Binding kinetic measurements of soluble RBD and RBD-PB NP on SPR sensor chips grafted with the ACE2 receptor. | | | |
| --- | --- | --- | --- |
| **ACE2-immobilized** | **Kd (nM) [mean ± SD]** | **kon ($M^{-1}s^{-1}$) [mean ± SD]** | **koff ($s^{-1}$) [mean ± SD]** |
| RBD-Fc monomer | 4.09 ± 2.29 | $3.11×10^5 ± 0.86×10^5$ | $1.99×10^{-3} ± 1.22×10^{-3}$ |
| Trimer RBD-PB NP | 1.26 ± 0.85 | $4.08×10^5 ± 1.37×10^5$ | $1.02×10^{-3} ± 1.12×10^{-3}$ |

throughout a 24-hour incubation in PBS containing 20% mouse serum, and the changes in fluorescence intensity of the acceptor dye were used to monitor the degradation rate. The stability of the NPs was measured from the FRET efficiency as calculated in Wei et al. (2013)[54]. Our FRET results showed that 21% of the bare PB DNA-NP were still intact after 24 hours in serum. Interestingly, PB modified with phosphorothioate-modified CpG strands, with 3-mer RBDs, as well as both 3-mer RBDs and CpGs, showed significantly increased stability in comparison to bare PB DNA-NP. Specifically, our results show that about 63% of the PB DNA-NP with only CpGs ($n = 3$ independent samples), 76% of PB DNA-NP with only 3-mer RBDs ($n = 3$ independent samples), and 68% of PB DNA-NP presenting 3-mer RBDs and CpGs together at opposite faces ($n = 3$ independent samples) remain intact after incubating them for 24 hours in serum (Fig. 3a). These results confirm that these DNA-NPs have the potential to remain intact after in vivo injection and until they reach their target within the body. More importantly, these results demonstrate that coating DNA-NPs with protein and modified nucleic acids (i.e., phosphorothioate groups) can contribute to increasing the stability against nucleases by shielding the DNA-NPs and blocking some of the 3'- and 5'-ends and by reducing the rate of degradation with the phosphorothioate groups. Thus, removing the need for complex chemical modifications proposed in literature[55] that can complicate the synthesis process and may lead to potential adverse reactions.

**Comparing binding affinity of free RBD monomers and RBD-trimers on DNA-NP.** Next, we used surface plasmon resonance (SPR) to validate the accessibility of the antigens on the surface of

the NPs by evaluating the binding kinetics of our PB DNA-NPs presenting 3-mer RBD in comparison with the free RBD. We modified the SPR gold sensor surfaces with the soluble ACE2 receptor. Immobilization of ACE2 was consistent across all the experiments performed with an average of 1540.3 ± 292.2 (mean ± SD) response units (RU) (Supplementary Fig. 16). Soluble RBD-Fc antigens at various concentrations (2 nM, 5 nM, 10 nM, and 25 nM) were injected over the ACE2 receptor and the equilibrium dissociation constant (Kd) between ACE2 and the free RBD was calculated at 4.47 nM (Fig. 3b). We used the PB DNA-NPs presenting the 3-mer RBDs at different concentrations (2 nM, 5 nM, 10 nM, and 25 nM equivalent RBD concentration) and determined a Kd of 1.35 nM (Fig. 3b). In each SPR experiment, a single-cycle kinetic measurement was performed, without regeneration of the surface, and with a long dissociation time before each injection (Supplementary Fig. 17). Table 2 summarizes the values (mean ± SD) associated to the binding association/dissociation rate constants (kon and koff, respectively) and the Kd calculated from the SPR curves. The Kd values determined in our studies are in the same range as previously published studies (Supplementary Table 8)[56–58].

**Immunizing BALB/c mice with DNA-NP nanovaccines elicits high antibody response.** After validating the availability of the RBD domain on the DNA-NPs via SPR, we prepared four different PB DNA-NP vaccine constructs including RBD-PB and RBD-PB-CpG to perform immunization assays with a BALB-C mouse model. Recent studies, including ongoing clinical trials with RBD, have used doses of RBD ranging from 1 to 90 µg[21,59–61]. Thus, we assessed two different RBD quantities

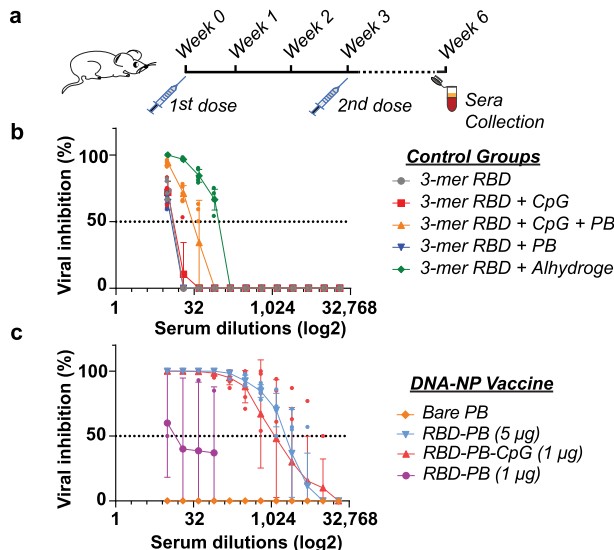

**Fig. 4 Immunization studies performed in BALB-C mice. a** Two doses of control samples and PB DNA origami nanovaccines (IM injection; 50 μl each) were administered with a three-week interval. Sera were collected at the end of the 6th week to assess antibody titers via PRNT and end-point dilution assays. **b** Virus neutralization assay was performed for the serum samples collected from control groups ($n = 5$ animals per study group) **c** and PB DNA origami nanovaccines. Among four different animals injected with nanovaccines ($n = 5$ animals per study group), significant viral inhibition was observed in mice immunized with RBD-PB (1 μg), RBD-PB (5 μg), and RBD-PB-CpG (1 μg). Data were shown as mean ± SD on graph. Two-way ANOVA was performed for the data shown in panel b and c to evaluate the statistical significance of data ($p < 0.001$).

(1 μg and 5 μg) presented on our PB DNA-NP nanovaccines. The quantity of RBD delivered was increased by injecting a higher concentration of NPs per dose. We also prepared five different control groups: 3-mer RBD (1 μg), 3-mer RBD (1 μg) + PB (unconjugated), 3-mer RBD (1 μg) + CpG (unconjugated), 3-mer RBD (1 μg) + PB + CpG (unconjugated), 3-mer RBD (1 μg) + Alhydrogel® to assess the effect of each component individually (Fig. 4). Two doses of each of the nine samples were injected intramuscularly (IM) to the mice with a three-week interval (Fig. 4a). Six weeks after the first injection, mice were euthanized, and the serum was collected to perform further tests.

The presence and neutralizing efficacy of antibodies in serum samples collected from animals injected with control (unconjugated) groups and DNA origami nanovaccine constructs were analyzed using a plaque reduction neutralization assay (PRNT) carried out by conventional methods[62,63]. Our PRNT results demonstrate that viral inhibition by serially diluted serum antibodies from control groups were found not to be as effective as the nanovaccine constructs (Fig. 4b). The comparisons among all control groups revealed that the serum antibodies collected from the animals injected with 3-mer RBD (1 μg) + PB + CpG (unconjugated) showed relatively high neutralization capacity followed by the even higher neutralization capacity of the antibodies from the 3-mer RBD (1 μg) + Alhydrogel® injected animals in a serum dilution less than 1:1024. The neutralization capacity of antibodies produced by the animals injected with 3-mer RBD (1 μg) alone, 3-mer RBD (1 μg) with PB (unconjugated), or 3-mer RBD (1 μg) with CpGs (unconjugated) were even lower with neutralization efficacy reduced drastically for serum dilutions superior to 1:32. As for DNA-NP nanovaccine constructs, RBD-PB NP (5 μg) and RBD-PB-CpG (1 μg) were both highly effective in eliciting a strong neutralizing antibody

response. The data showed strong inhibition of viral infectivity from the Vero monolayer at >1:1024 dilution, whereas PB alone had no effect on viral inhibition and PB with 1 μg RBD had limited efficacy (Fig. 4c). Given that the outcomes of virus neutralization by serum antibodies from control samples and nanovaccines, all control samples tested did not induce a response that was as strong as the nanovaccine constructs.

**DNA-NP nanovaccines protect against aerosol challenge with live SARS-CoV-2.** The initial assessment of immunogenicity also suggested that no adverse events were observed in the vaccinated animals under the dosage regimen employed. Adverse events were all reported based on the changes in physical appearance, mobility, attitude, body features (Supplementary Table 9 and Supplementary Table 10) of mice upon the administration of control or vaccine samples and all changes were recorded according to the given scores in 'Animal Study Clinical Monitoring Chart'. As a next step, we evaluated if the immunization regimen could confer protection in the face of a lethal challenge by SARS-CoV-2 in the K18-ACE2 mouse model (transgenic mice expressing the ACE2 receptor)[64–66]. We injected the aforementioned five different control groups as well as five different DNA-NP nanovaccine constructs (PB, PB-CpG, 1 μg RBD-PB, 5 μg RBD-PB, and 1 μg RBD-PB-CpG). After injecting two doses of the control groups and DNA-NP nanovaccine constructs with a three-week interval, the mice were exposed to the live SARS-CoV-2 (Isolate Italy, INMI1)[67] via intranasal route at a dose of $5 \times 10^4$ plaque-forming unit (pfu) per mouse (Fig. 5a). The survival rate of each animal was evaluated for 14 days, along with daily monitoring of vaccinated animals weight loss (Fig. 5b, c). The animals injected with the unconjugated samples showed a decrease in the survival rate after the viral challenge. For instance, we observed a mortality of 100% for the animals injected with 3-mer RBD (1 μg) + PB (unconjugated); 80% mortality for those injected with 3-mer RBD (1 μg) + CpG (unconjugated); 60% mortality with those injected with RBD (1 μg) or RBD (1 μg) + PB + CpG (unconjugated). Interestingly the animals injected with RBD (1 μg) + Alhydrogel® showed the lower mortality rate of all the control samples tested with only 40% mortality. For the nanovaccine group, at Day 7 after viral exposure, mice immunized with Bare PB, RBD-PB (1 μg), and PB-CpG showed a nearly 30% weight loss, whereas mice immunized with RBD-PB (5 μg) and RBD-PB-CpG (1 μg) did not show weight loss. At Day 7, the groups injected with Bare PB, RBD-PB (1 μg), and PB-CpG had a drop in the survival rate with 60% mortality for PB-CpG and 80% mortality for Bare PB and RBD-PB (1 μg). At Day 14, animals vaccinated with 5 μg dose of RBD-PB showed only 40% mortality. At Day 14, only two mice injected with PB and PB-CpG survived. Interestingly, the PB DNA-NP nanovaccine construct prepared with CpG plus 1 μg dose of RBD showed no mortality and also no body weight loss over the 14-day period, thus confirming the protection provided by our nanovaccine construct. Recent studies have shown that a CpG 1018 adjuvant is effective in the induction of neutralizing antibodies and Th1-biased cell responses against SARS-CoV-2, which seems to be confirmed in our study[52,53,68]. According to our results, administering PB with CpG and 1 μg of RBD together enhanced immunity against the virus. Furthermore, using CpG clearly improved immunization and allowed the use of lower antigen quantities to trigger a specific and strong immune response.

Furthermore, to better understand the type of response triggered by our vaccine, we assessed the RBD-specific antibodies (i.e., IgM, IgG, IgA) produced in the immunized mice. Indeed IgG and IgM are critical for immune protection against SARS-CoV-2 through humoral immunity and IgA are important for mucosal

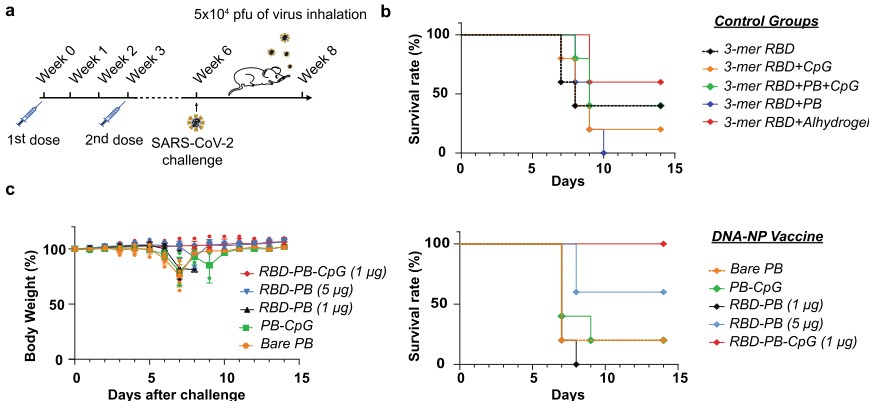

**Fig. 5 Challenge studies with SARS-CoV-2 virus. a** Animals ($n = 5$ animals per study group) immunized with two doses of control groups and PB DNA origami nanovaccine samples were subjected to challenge with SARS-CoV-2 via intranasal route of infection at Week 6. **b** Survival rate and **c** body weight changes (mean ± SD) were monitored for 14 days after intranasal viral challenge. After 14 days, mice administered with the 3-mer RBD (1 μg) + Alhydrogel® survived with the highest survival rate of 60% among the other control groups. Among the vaccinated mice with DNA origami nanovaccines, 60% of mice survived from those vaccinated with RBD-PB (5 μg). Only one animal survived among mice vaccinated with bare PB and one animal vaccinated with PB-CpG. Similarly, only one animal which was injected with the 3-mer RBD (1 μg) + CpG (unconjugated) sample succeeded to survive. All mice injected with the 3-mer RBD (1 μg) + PB (unconjugated) sample and with RBD-PB (1 μg) vaccine candidate succumbed at Day 10 and Day 8, respectively. There was no weight loss and death observed among mice vaccinated with (1 μg) RBD-PB-CpG. Weight loss of post-challenged mice were compared with two-way ANOVA ($p < 0.001$).

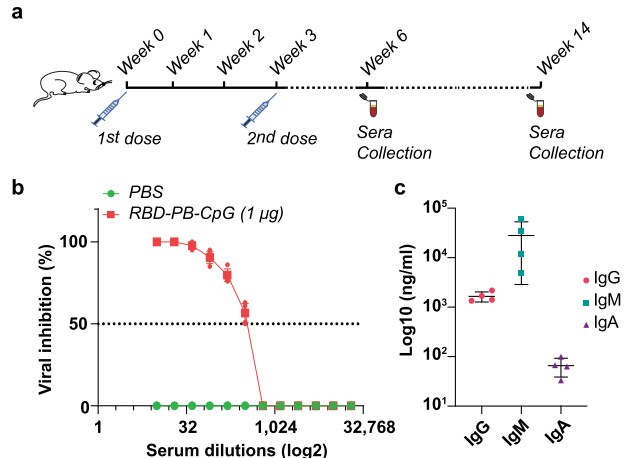

**Fig. 6 Durability of the immune response triggered by our nanovaccine. a** Serum antibodies were collected 2 months after the second dose of immunization. **b** Neutralization efficacy was tested for mice injected with a DNA-NP vaccine construct (1 μg RBD-PB-CpG) or PBS ($p < 0.001$) ($n = 5$ animals per study group). Two-way ANOVA was performed for statistical analysis of viral inhibition assay. **c** RBD-specific IgG, IgM, and IgA antibody titers were determined via ELISA assay. The Tukey's multiple comparison test was performed and showed that there is no significant difference between the means of antibody concentrations (IgG, IgM, and IgA) ($p > 0.05$) ($n = 4$ animals).

immunity, which is a key component of the response against SARS-CoV-2[69]. We calculated the titers for each type of antibody via ELISA assay (Supplementary Fig. 18) and evaluate the durability of the antibody responses in an extended period (2 months after the boost dose) using blood serum of animals immunized with 1 μg RBD-PB-CpG vaccine construct. Strikingly, 50% virus neutralization was observed with a serum dilution between 1:256 and 1:512, slightly less than what was obtained for the initial results 15 days after the second injection (Fig. 6). The concentration of RBD-specific IgG was found to be about 1,655.5 ng/ml, IgM was about 28,031 ng/ml, and IgA was about

66.0 ng/ml. The cross-reactivity was not detected based on the tests applied by the manufacturer. In previous human clinical studies with mRNA vaccines applied on human[70,71], similar antibody levels (ng/ml) were also reported. Overall, since dose differences and responses between different species can result in different outcomes in immunization, it is difficult to perform direct comparison between the response triggered by mRNA vaccines and our strategy.

## Conclusions

In this study, we designed a 3D DNA origami PB and displayed 10 copies of a SARS-CoV-2 RBD trimer along with 10 CpG adjuvants on the two opposite faces of the NP. The viral challenge experiments showed that even small antigen doses delivered by the DNA-NPs are sufficient to provide protection against the virus when administered along with adjuvants. Interestingly even two months after the second dose, the animals demonstrate robust immunity with highly specific antibodies against RBD. Altogether, our results clearly demonstrate the potential of this strategy to develop a vaccine and highlight the importance of testing the spatial organization and stoichiometry of antigens to maximize the cellular response. Moreover, our DNA-NPs could be used to present multiple types of immunogens simultaneously to develop a broad-spectrum vaccine targeting multiple viral strains.

In addition to paving the way toward using DNA-NPs as the potential next generation of nanoparticle-based vaccine, our strategy demonstrate that rational design of DNA-NP-based vaccine will be key to reduce the high cost associated with this technology. Indeed, we have shown that properly organizing antigens and co-presenting adjuvants contribute to a significant decrease in the required quantity of antigen to induce strong immunization and protection in a mouse model. The controlled organization of the antigens and adjuvants simultaneously in the DNA-NP will greatly reduce the overall cost per dose of this DNA-NP-based vaccine strategy. However, it is important to note that the strategy for large-scale production of ssDNA scaffold are still required to reduce the cost of synthesis of these DNA-NPs. Additionally, updated methodologies to optimize particle

purification in scaled-up production will be beneficial to reduce the number of purification steps and increase the final production yield. Altogether, the results presented in this manuscript demonstrate the potential and the robustness of this vaccine strategy and pave the way toward rational design of vaccine nanoparticles that could be used to rapidly respond to future viral threats.

## Methods

**Reagents, consumables, and kits.** The oligonucleotides used as primers in the asymmetric polymerase chain reaction (aPCR) and as staple strands for the folding of PB NPs, along with the CpG ODN 1018 adjuvant were purchased from Integrated DNA Technologies (IDT). Fluorescently labelled oligonucleotides used in FRET assay, FAM (donor) and TAMRA (acceptor), were also procured by IDT. The deoxynucleotide triphosphates (dNTPs) mix (cat. no: N0447L), the Quick Load Purple 1 kb plus DNA ladder (cat. no: N0550S), and the M13mp18 circular single-stranded DNA template (cat. no: N04040S) was obtained from New England Biolabs (NEB). The AccuStart Taq DNA polymerase HiFi enzyme was obtained from Quanta Biosciences. Low melt agarose (cat no: 89133-104) was provided by IBI scientific. The Zymoclean Gel DNA recovery kit was provided by Zymo Research (cat. no: D4008). The 10 kDa MWCO (cat. no: UFC5010) and 100 kDa MWCO (cat. no: UFC5100) AmiconUltra 0.5 centrifugal filters, the 11- mercaptoundecanoic acid (CAS no: 71310-21-9) and L-Tryptophan (CAS no: 73-22-3) were obtained from Sigma Aldrich. The Proteus X-Spinner 300 kDA MWCO filter was purchased from Protein Ark. The RBD, Fc (cat. no: SPD-C5255) and the biotinylated human ACE2/ACEH, His, Avitag (cat. no: AC2-H82E6) were provided by ACRO Biosystems. The cys-protein G was purchased from Prospec (cat. no: pro-1238). The PNA-Maleimide was procured by PNA Bio. The Gold sensor chips for SPR experiments were obtained from Nicoya Lifesciences (cat. no: SEN-AU-100-10). Greiner 384-well polystyrene flat bottom microplate was purchased from Cellvis (cat no: P384-1.5H-N). PageBlue Protein Staining Solution (cat. no: 24620), 1% crystal violet (cat. no: C581-25) and 20% ethanol solution (cat. no: BP2818-4), formaldehyde (cat. no: F79p-4) and 0.6% agarose (cat. no: 16500100) were purchased from Thermo Fischer Scientific. 1 mM sodium pyruvate was purchased from VWR (cat. no: 45000-710). Mouse anti-2019-nCoV(S) ELISA kits (IgA kit cat. no: DEIASL617; IgG kit cat. no: DEIASL618; IgM kit cat. no: DEIASL619) were purchased from Creative Diagnostics®. Alhydrogel® adjuvant 2% was procured by InvivoGen (CAS no: 21645-51-2). Cy5-NHS linker was provided by Nanocs (cat. no: S5-1, 2).

**Cell lines and animals.** Vero cells were purchased from ATCC. BALB-C mice were purchased from Jackson Laboratories. K18-Ace2 (B6.Cg-Tg(K18-ACE2)2Prlmn/J) mice were purchased from Jackson Laboratory (Stock no: 034860).

**Pentagonal bipyramid (PB) scaffold production.** The PB single-stranded DNA (ssDNA) scaffold was synthesized via aPCR using the protocol described in Veneziano et. al. in 2018[48]. Briefly, 50 µl reaction mix was prepared with 25 ng of M13mp18 template, the PB primer set in a 1:50 molar ratio (1 µM forward primer, 20 nM reverse primer), 1X HiFi buffer (provided by Quanta Biosciences) supplemented with 2 mM of Magnesium Sulfate (MgSO₄), 0.2 mM of the dNTPs mix, and 1.25 U of AccuStart Taq DNA Polymerase HiFi. The aPCR cycle were performed in a Bio-Rad T100 Thermal Cycler as follow: activation at 94 °C for 1 min followed by 30-40 cycles of 94 °C for 20 s, 55 °C for 30 s, and the amplification step at 68 °C for 2 min. Low melt agarose (1.2%) preloaded with ethidium bromide was used to visualize and purify the ssDNA scaffolds. The ssDNA band was cut from the gel and purified using the Zymoclean Gel DNA recovery kit following the vendor instructions. The ssDNA scaffold was quantified using Nanodrop and the purity was assessed via gel electrophoresis.

**Folding of PB nanoparticle (NP).** The ssDNA scaffold was folded into PB NP using a 1:10 molar ratio of scaffold *vs* staple strands. The folding reaction was performed via an overnight annealing in TAE-Mg²⁺ buffer (40 mM Tris, 20 mM acetic acid, 2 mM EDTA, and 12 mM MgCl₂, pH 8.0) from 95 °C to 4 °C as previously described in Veneziano et al[36]. PB folding was validated via agarose gel electrophoresis and imaged via Azure™ c150 imager (Azure Biosystems, Inc.). The yield was quantified using Image J.

**Conjugation of PNA strand to the Cys-Protein G (PG)-Cysteine.** PG with N-terminal cysteine was reduced with 10-fold molar excess of tris(2-carboxyethyl) phosphine (TCEP) at room temperature for 15 min. After 15 min, PG was filtered via 10 kDa Amicon Ultra-0.5 centrifugal filter to remove excess of TCEP and reacted overnight with 3-fold molar excess of PNA-Maleimide at 4 °C. PG-PNA purification was achieved with 11 filtration steps using 10 kDa Amicon Ultra-0.5 centrifugal filter to remove the unbound PNA. [The sequence of PNA with maleimide is: 'SMCC-GGK-cagtccagt-K', which is composed of amino acid residues (shown as uppercase) and bases (shown as lowercase)].

**Reconstitution of the RBD trimer immunogen.** To assemble 3-mer RBD, purified PG-PNA was mixed with 5-fold molar excess of RBD-Fc and incubated in 1X phosphate buffer saline (PBS) at 37 °C for 1.5 h and purified with Amicon Ultra filter 100 kDa to remove the excess of monovalent RBD-Fc using at least three centrifugation steps.

**Functionalization of the PB nanoparticle with the RBD trimers and the CpG strands.** Attachment of RBD-PG-PNA to PB NP was performed via hybridization to DNA overhangs displayed by staple strands on the DNA-NP at 37 °C for 1.5 h. The molar ratio between PB NP and RBD-PG-PNA was 1:2, respectively. For conjugation with CpG, a 10-fold molar excess of CpG ODN 1018 was used and the reaction was done at 37 °C for 1.5 hr. Reaction product was filtered at least 3 times from excess amount of CpG and PG-RBD protein complex using centrifugal filter (300 kDa MWCO). Purified RBD-PB-CpG was kept at 4 °C before further use. (Sequence of CpG 1018 with linker complementary to specific overhangs on PB is the following:

5'-T\*G\*A\*C\*T\*G\*T\*G\*A\*A\*C\*G\*T\*T\*C\*G\*A\*G\*A\*T\*G\*A ACTTCATGGTCCTAACTT-3'. (\* indicates phosphorothioate linkage and the underlined sequence indicates the linker sequence for hybridization to the overhangs).

**Preparation of control groups.** All control groups were composed of the unconjugated mix of separate components from the DNA-NP vaccine formulations wherein incubation conditions necessary to achieve conjugation were skipped. Control groups include the injection of 50 µl placebo (1X PBS), free 3-mer RBD (1 µg), 3-mer RBD (1 µg) + PB (26.7 nM) (unconjugated), 3-mer RBD (1 µg) + CpG ODN 1018 (0.27 µM) (unconjugated), 3-mer RBD (1 µg) + PB (26.7 nM) + CpG ODN 1018 (0.27 µM) (unconjugated), 3-mer RBD (1 µg) and Alhydrogel® per animals for the immunization and challenge part of the study as comparisons. The concentration of CpG ODN 1018 used in control samples was similar to the theoretical concentration obtained on DNA-NP with 10-overhang. As for RBD + wet gel alum (Alhydrogel®), 1 µg of RBD was mixed with Alhydrogel® in a 1.1:1 volume ratio (1:20 mass ratio) for 50 µl injection per animal, at least 30 minutes before injection to promote sufficient adsorption of RBD on Alhydrogel®. Particularly, based on the previous studies referring the adsorption capacity of aluminum hydroxide-containing adjuvants on proteins[72,73], besides the information provided from vendor, led us to use this volume ratio between protein and adjuvant.

**Atomic force microscopy imaging.** DNA-NPs (i.e., bare PB, PG-PB, and RBD-PB) were diluted to approximately 2 nM in filtered 0.5X TBE buffer (50 mM Tris Base, 50 mM Boric acid, 1 mM EDTA, pH 8.3) supplemented with 12.5 mM MgCl₂ and 15 µl of each sample were deposited onto freshly cleaved mica disk for 5 min incubation. The mica surface was then rinsed of any unabsorbed DNA and proteins by twice depositing 100 µl of the filtered buffer onto the mica and wicking the solution from the mica. For imaging, 100 µl of buffer was supplemented with 5 mM NiCl₂ and added to the mica surface. Images were taken with areas of 1 × 1 and 2 × 2 µm² with 1000 pts/line and 1000 lines/scan with scan rates of 8 and 4 Hz, respectively. Imaging was performed in the fluid using USC-F0.3-k0.3 cantilever tips (NanoWorld) on a JPK instruments NanoWizard 4 fast-scan AFM. Images were processed using open-source software Gwyddion.

**Defining the stoichiometry of proteins coating DNA-NP.** To elucidate the total number of proteins attached onto the surface of PB DNA-NPs, we executed individual analysis for each protein on NP. These analyses were based on fluorescence emission spectroscopy, wherein we measured either the fluorescence emission of the amino acid (tryptophan) or cyanine dye (Cy5) or quantified the ratio between proteins on NP using the mass-spectrometry.

**Measuring intrinsic tryptophan fluorescence intensity of PG on NP.** Utilizing the optical properties of tryptophan due to the aromatic ring, we established a standard curve based on the fluorescence emission of varying concentrations of tryptophan in PBS (concentration range between 0 to 20 µM). Particularly, to distinguish the emission of tyrosine from tryptophan, the excitation wavelength was set to 295 nm. The emission wavelength was set from 325 nm to 500 nm. DNA-NP folded by the staples with 10 overhangs reacted with 2-fold molar excess of PG-PNA and purified from excess PG-PNA using Amicon Filter (100 kDa MWCO). Based on NP concentration, we estimated the protein concentration on the NP. By using the tryptophan standard curve, free PG-PNA and NP with PG-PNA which were assumed having same concentration of PG were calculated.

**Assessment of RBD stoichiometry on NP via measuring Cy5 emission.** The labeling reaction of RBD with Cy5-NHS was performed overnight at 4 °C. Cy5-NHS was mixed with RBD (at a ratio of 1:10) in sodium bicarbonate (NaHCO₃) pH 8.4. Next day, the excess Cy5-NHS was removed from the solution via spin-filtration using Amicon Filter. In the second step, the Cy5-NHS-labeled RBD was attached to the NP through incubating at 37 °C. The standard curve (concentration range between 0 to 10 µM) was established according to the emission from Cy5 dye

**Table 3 The gradient of solvents performed in liquid chromatography.**

| Time (min) | A% | B% | Flow rate (ml/min) |
|---|---|---|---|
| 0 | 100 | 0 | 0.3 |
| 2 | 100 | 0 | 0.3 |
| 45 | 70 | 30 | 0.3 |
| 50 | 0 | 100 | 0.3 |
| 55 | 0 | 100 | 0.3 |
| 55.1 | 100 | 0 | 0.3 |
| 60 | 100 | 0 | 0.3 |

on RBD, in which the excitation was set at 590 nm and the emission spectrum range was arranged between 640 nm and 700 nm.

**Mass spectrometry for quantifying PG to RBD ratio on NP**. A sample of NPs as well as a serial dilution of the protein standards (RBD and Protein G) were dissolved in a total volume of 100 μl of 50 mM ammonium bicarbonate buffer (pH=7.8). Dithiothreitol (DTT) was added to reduce disulfide bridges to a total concentration of 5 mM. The reaction mixture was incubated at 56 °C for 60 min. The protein mixture was alkylated using iodoacetamide to a total concentration of 5 mM at 37 °C for 30 min under dark conditions. The reaction mixture was left to cool down to room temperature and quenched with DTT solution in 50 mM ammonium bicarbonate. The protein content was digested using Trypsin Gold, MS-grade (Promega, V5280) at a ratio of 1 to 100 (protein to trypsin) by mass and incubated overnight at 37 °C. The enzyme was denatured by heating the reaction mixture to 90 °C for 10 min. The samples were desalted on a C18 spin column (Nest Group, cat. no. HEM S18V) and peptides were eluted using a mixture of 80% LC/MS (liquid chromatography/mass spectrometry) grade acetonitrile and 20% LC/MS grade water acidified with 0.1% formic acid. The solvent was evaporated under vacuum using a SpeedVac then reconstituted with LC/MS grade water acidified with 0.1% formic acid containing the internal standard.

The reconstituted peptide mixture was analyzed on a Sciex QTRAP 4500 mass spectrometer equipped with a Shimadzu Prominence UFLC XR System, whereas it was separated on Xterra MS C18 column, 3.5 μm, 2.1 × 150 mm, PN: 186000408. The column temperature was kept at 23 °C. Solvent A was 100% LC/MS grade water with 0.1% formic acid and solvent B was 100% LC/MS grade acetonitrile with 0.1% formic acid. The flow rate was constant throughout the method and was set to 0.3 ml/min. The gradient is presented in Table 3.

The mass spectrometry method was set to a positive MRM mode with a cycle time of 0.9450 seconds. All transition had a dwell time of 40 msec and the collision energy (CE) was set to 30 volts. The Entrance Potential (EP) was set to 10 volts and the Declustering Potential (DP) was set to 50 volts. The Collision Cell Exit Potential (CXP) was set to 14 volts. Data was analyzed using Analyst 1.7 software.

**Assessment of CpG coverage yield on NP via measuring fluorescein emission**. Fluorescence-intensity-based quantitate measurement was applied to ascertain the approximate number of CpG hybridization over DNA-NP. The standard curve was obtained based on the emission spectra of fluorescein in varying concentration of fluorescein-labelled CpGs (0 to 10 μM). The excitation wavelength was set at 475 nm and emission wavelength was set between 500 to 700 nm with a maximum emission peak at 520 nm. According to the fluorescence intensity of a reference fluor-CpG ODN concentration, the number of modifications over DNA-NP by CpG ODN was estimated for DNA-NP samples modified with the fluorescently labelled CpGs as described earlier in the method section.

**Förster resonance energy transfer (FRET)-based stability assay**. PB NP (bare PB, PB-CpG, RBD-PB, and RBD-PB-CpG) were modified with two FRET pairs (Donor: fluorescein [FAM] and acceptor: TAMRA). The FRET NPs (2.4 pmol) were incubated in 20% (v/v) mouse serum to assess the relative stability ratio throughout a 24-hour period. The fluorescence measurements were performed in a Microplate reader (Tecan Safire2) with an excitation at a wavelength of 455 nm (20 nm bandwidth) and the emission spectra collected from 500 nm to 700 nm (20 nm bandwidth). The samples were loaded into Greiner 384-well polystyrene flat bottom black microplate. The following equations were used to calculate the FRET efficiency and determine the rate of degradation. We used the change in the fluorescence intensity of donor dye over time according to the method proposed by Wei et al.[54] According to Eqs. 1 and 2 (see Eqs. (1) and (2), respectively) below:,

$$Fret\ efficiency\ (E) = \frac{(ID - IDA)}{ID} \tag{1}$$

$$Assembled\ particles\ fraction\ (\theta) = \frac{(E - E\min)}{(E\max - E\min)} \tag{2}$$

where ID represents the intensity of donor dye from only donor bearing PB NPs

and IDA represents the intensity of donor dye from donor-acceptor bearing PB NPs.

**Binding kinetics measurement of the RBD on ACE2 receptors via surface plasmon resonance**. Binding association and dissociation kinetics between ACE2 and RBD-Fc or PB-RBD were measured with an Open Surface Plasmon Resonance (Open SPR) instrument provided by Nicoya Lifesciences. Gold sensor chips were first prepared by immersion in 11-mercaptoundecanoic acid and kept at room temperature for 48 h. Prior to the SPR measurements, the amino gold surface was biotinylated by incubation with 100 μl EDC/NHS mix from a stock of 200 mM EDC and 50 mM NHS for 3 min at room temperature, which was followed after thorough rinsing with water by an incubation with BSA-biotinylated (0.5 mg/ml) in 10 mM HEPES, 150 mM NaCl buffer at pH 7.2 for another 3 min at room temperature. The sensor chip was then loaded in the SPR device and a streptavidin solution (1 mg/ml) in running buffer was flowed over biotinylated gold surface chip at a flow rate of 20 μl/min. The running buffer used was 10 mM HEPES, 150 mM NaCl buffer at pH 7.2 and supplemented with 0.1 mg/ml of BSA and 0.05% (v/v) Tween-20 as recommended by the vendor. ACE2-biotinylated was immobilized on the streptavidin surface by injection of 150 μL at a concentration of 20 μg/ml and at a flow rate of 10 μl/min. Through performing single-cycle kinetics, dissociation times were monitored for at least 900 s before each injection of analytes. We used concentrations of 25 nM, 10 nM, 5 nM, and 2 nM of RBD-Fc monomers and equivalent RBD concentration with the RBD-PB NPs. RBD and RBD-PB were analyzed in separate experiments. Measurements for each concentration were performed at least in triplicate. Nicoya OpenSPR instrument was used to fit the kinetic data using a 1:1 Langmuir binding model.

**Size measurement of PB NPs**. The hydrodynamic diameter of the different PB NPs used was determined via dynamic light scattering (DLS) with a NanoZetaSizer (Malvern Instruments, Ltd). Technical replicates (n = 3) of 100 nM PB NP (10-mer), PB NP (20-mer), PG-PB NP, and RBD-PB NP were performed in 1X phosphate buffer saline (PBS) at 25 °C. Intensity-weighted Z-average diameter for each PB NPs were reported along with their polydispersity index (PDI).

**Animal immunization**. We prepared solution of 50 μl of four different PB DNA-NP nanovaccine constructs: PB alone, RBD-PB (1 μg and 5 μg doses), RBD-PB-CpG (1 μg dose) and five different control groups: free 3-mer RBD (1 μg), 3-mer RBD (1 μg) + CpG (unconjugated), 3-mer RBD (1 μg) + PB (unconjugated), 3-mer RBD (1 μg) + PB + CpG (unconjugated), 3-mer RBD (1 μg) + Alhydrogel® that were administered to 6 to 8 weeks old female BALB/c mice via intramuscular (IM) injection in the right caudal thigh muscle. Two injections were done with three-week intervals. Animals were examined daily to assess indications of distress according to the parameters defined by 'Animal Study Clinical Monitoring Chart' approved by the GMU IACUC. After three weeks from the second injection and further 14 weeks for extensive study on RBD-PB-CpG (1 μg dose) and placebo, the mice were euthanized. Serum separated from blood collected from submandibular vein were examined via plaque reduction neutralization assay (PRNT).

**Plaque reduction neutralization assay (PRNT)**. Neutralization antibody titers from each control group and PB DNA-NP nanovaccines were determined through virus inhibition with PRNT assays. Sera collected three weeks after the second injection was two-fold serially diluted in four steps starting with 1:10 dilution in Dulbecco's Modified Eagle Medium (DMEM) supplemented with 5% fetal bovine serum, 1% L-glutamate, 20 U/ml penicillin, and 20 ug/ml streptomycin. Each sera dilutions were mixed with 100 pfu of virus (SARS-CoV-2, Isolate Italy-INMI1) and serum-virus mix was incubated at 37 °C with 5% CO2 for 1 h. After incubation, mixture was inoculated into confluent layer of Vero cells in a 12-well plate and incubated at 37 °C with 5% CO2 for 1 h. After an hour, 0.6% agarose (Thermo-Fisher, 16500100) containing Eagles's Minimum Essential Medium (EMEM–without phenol red) supplemented with 5% FBE, non-essential amino acids, 1 mM sodium pyruvate (VWR, 45000-710, Dixon, CA, USA), 2 mM L-glutamine, 20 U/mL penicillin, 20 μg/mL streptomycin was added to each well in a 1:1 volume ratio. The cells were then incubated at 37 °C with 5% CO2 for 48 h. After the incubation period, cells were fixed with 10% formaldehyde (Fisher Scientific, F79p-4) for 1 h. The formaldehyde/agarose plugs were removed after 1 h. Cells were then washed with deionized water, then stained with 1% crystal violet (FisherSci, C581-25) and 20% ethanol solution (FisherSci, BP2818-4). Plaques were counted and analyzed in a plot showing dilution versus pfu values.

**Viral inhalational challenge with SARS-CoV-2 strain Isolate Italy-INMI1**. 6-8-week-old male K18-Ace2 (B6.Cg-Tg(K18-ACE2)2Prlmn/J) mice were subjected to challenge studies with five different control groups aforementioned and PB DNA-NP nanovaccine constructs, which were PB alone, 1 μg and 5 μg doses of 3-mer RBD-PB, 1 μg dose of 3-mer RBD-PB-CpG ODN 1018, and CpG ODN1018-PB. 50 μl of each construct was injected via IM route from the right caudal thigh for initial immunization. After three weeks, a second injection was performed. Animals were monitored daily to observe distress parameters in ABSL2 vivarium. At the end of 6th week, mice were moved to ABSL3 vivarium and infected with SARS-CoV-2 (Isolate Italy-INMI1) at a dose of 5 × 10⁴ pfu through intranasal route. Each

animal injected with different control groups and nanovaccine constructs were monitored according to individual distress parameters and the survival rate was recorded by 'Animal Study Clinical Monitoring Chart' provided by Animal Care Services.

**Enzyme-linked immune-sorbent assay (ELISA)**. ELISA kit for RBD-specific serum IgG, IgM, and IgA detection were used as described in the protocol provided by Creative Diagnostics®. Serum samples were diluted in sample dilution buffer 1:100 for IgG, 1:100 for IgA, and at least 1:1000 for IgM detection. Wells were washed twice with 1X washing buffer and incubated with 50 μl standard samples (concentration range between 1.563 to 100 ng/ml for IgG, 0.781 to 50 ng/ml for IgM, and 0.156 to 10 ng/ml for IgA, with a dilution factor of 2) diluted serum samples for 30 min in 37° degree. Solutions in the wells were discarded after incubation and wells were washed triple before the incubation with 50 μl of HRP (Horseradish peroxidase)-secondary antibody (1:100 dilution) for 30 min. After wells were washed five times with washing buffer, TMB substrate were added into each well for 15 min incubation until the color changes. After 15 min, stop solution was added and absorbance of each well were measured at OD 450 nm. Standard curves (four parametric logistic curves) for determination of IgG, IgM, and IgA concentrations were prepared using MyAssays online data analysis tool[74].

**Ethical statement**. All animal studies carried out for these studies were in accordance with recommendations of the Institutional Animal Care and Use Committee (IACUC protocol #0399) at GMU.

**Statistics and reproducibility**. Stability data of PB nanoparticles and viral inhibition were reported as mean and standard deviation as shown on the graphs. The p-value was calculated to be less than 0.001 by two-way ANOVA and Tukey's post-hoc test was applied as multiple comparisons test for the stability of each NPs group using R studio version 1.2.5033. Statistical significance value was determined for viral inhibition data via two-way ANOVA, where the p-value was calculated less than 0.001 using R studio version 1.2.5033 and JASP 0.16.4.0. Body weight changes ($n = 5$ animals per study group) of five animals 14 days after challenge was analyzed with two-way ANOVA. The p-value was calculated to be less than 0.001. One-way ANOVA was used for the statistical analysis of each RBD-specific antibody response (IgG, IgM, and IgA). All statistical examinations were performed using R studio.

**Reporting summary**. Further information on research design is available in the Nature Portfolio Reporting Summary linked to this article.

## Data availability
The source data behind the graphs in this article can be found in Supplementary Data 1. Other data can be made available upon reasonable request to the corresponding authors.

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

## Acknowledgements

This study was partly funded by the DOD award number W81XWH2010054 and by a COS/VSE Collaborative Seed Fund award. We thank for the support of E.O received by the Turkish Ministry of National Education via the 'MoNE Scholarship'. I.L.M. acknowledges the Office of Naval Research and the U.S. Naval Research Laboratory. D.M. was supported by the National Institute of Biomedical Imaging and Bioengineering of the National Institutes of Health under Award Number R00EB030013-03. The content is solely the responsibility of the authors and does not necessarily represent the official views of the National Institutes of Health. We also thank Amanda Graf for her contribution to the production of DNA origami nanoparticle scaffold.

## Author contributions

R.V. conceptualized and designed the in vitro studies. A.N., F.A., and R.V. conceptualized and designed the in vivo studies. E.O. performed DNA-NP vaccine design, assembly, and characterization as well as all in vitro experiments. F.A. and K.H. performed the animal experiments. F.A. and K.H. collected animal data. M.G performed mass spectrometry experiments and analyzed the MS data. I.L.M., C.G., and D.M. conducted AFM imaging and AFM image analyses. E.O., F.A., and K.H. performed statistical analysis. E.O and R.V wrote the manuscript. All authors reviewed and edited the manuscript.

## Competing interests

Dr. Veneziano is listed as an inventor on submitted patents related to this work. All other authors declare no competing interests.

## Additional information

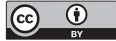

