## [Peer Review File · Communications Biology]

Reviewers' comments:

Reviewer #1 (Remarks to the Author):

The manuscript by Esra Oktay et al. describes a novel SARS-CoV2 vaccine formulation that utilizes protein antigen (RBD of SARSCoV2 virus), an adjuvant (CpG oligonucleotide), and a nanocarrier (DNA-origami nanoparticle). The authors present a physicochemical characterization of the DNA origami-based vaccines and efficacy studies in a mouse model.

Before publication, I suggest the authors consider several additional controls and questions summarized below:

1. The efficacy studies presented in Figures 3 and 4 would benefit from adding three controls - RBD-CpG conjugate, unconjugated RBD+CpG mixture, and RBD+PB+CpG unconjugated mixture of three components. These controls would help answer two very important questions about the need for a) the PB-component and b) conjugation of individual components for vaccine efficacy.
2. Another important control is RBD+clinical grade adjuvant, for example, Alum
3. How does the efficacy of DNA-origami-based formulation compare to that of LNP-mRNA formulations that are already in clinical use?
4. How sustainable is the immune response? How long does the protection last? The main disadvantage of LNP-mRNA vaccines is their short-term action which requires a boost after five months. Does the PB-RBD-CpG vaccine overcome this limitation?
5. Does this formulation induce RBD-specific antibodies (IgM, IgG, IgA), and at what titer? What is the specificity of these antibodies? Does it induce RBD-specific cytotoxic T-cell response? Is there any cross-reactivity of the induced antigen-specific response to normal antigens?
6. What is the role of the PB component? Does it have an adjuvant activity, deliver CpG and RBD to APCs, or both? How does the PB deliver antigen and adjuvant to APCs?
7. Is the PB-component immunogenic, i.e., does it induce DNA-origami specific antibody response? Can DNA origami break tolerance and induce DNA-specific antibodies? Answering these questions would help to eliminate autoimmunity concerns.

Reviewer #2 (Remarks to the Author):

This manuscript developed a DNA origami nanoparticle-based vaccine for Sars-Cov-2 by co-delivering RBD antigen and CpG adjuvant. The authors described the vaccine fabrication methods thoroughly and proved the vaccine efficacy in health mice immunization and mouse virus challenge study.

The overall impression of this manuscript: 1). It's well written in detail and easy to understand; 2). It provided mouse studies to evaluate the vaccine efficacy. 3). However, there are several major concerns to be addressed before the authors can make the conclusion that the nanoparticle really helps.

Majors:

1. Antigen and adjuvant are essential to stimulate strong immune responses. The authors also cited the FDA-approved vaccine HBsAg-CpG in 2018. However, the researchers in this work didn't include proper control in the animal study to compare with RBD and CpG given freely (we call bolus vaccine control). That is the key to show the advantage of the DNA origami nanoparticle. Otherwise, why would people use complicated and expensive DNA origami? Both figure 3 and figure 4 should include this control.
2. TEM images should be included to show the robust vaccine fabrication. I don't see any TEM results in this manuscript, which is not acceptable for DNA origami studies. In figure 2a, when 5 times more of RBD was used to incubate with PB, the band shift looks to me like aggregations. This can be caused by protein fabrication with DNA structures. TEM images should be able to tell if higher-order aggregation was formed.
3. I don't see a vaccine purification method mentioned in the manuscript. I also don't see the determination of component conjugation efficiency (e.g. CpG and RBD). It's not like 10 overhangs included, then 10 molecules will be conjugated. This can be determined by DNase digestion and

running denature page for CpG or SDS page for protein. The purification and conjugation efficiency should be essential to vaccine fabrication.

Minors:

1. Line 120, the second sentence of the Fig. 2a legend is not clear to me.
2. Line 216, the authors mentioned there are no adverse events. But I don't see any supporting data for this.
3. Line 170, it's interesting the authors mentioned: "coating DNA-NPs with protein and modified nucleic acids can participate in increasing the stability against nucleases without the need for complex chemical modifications as seen in extant literature". Maybe the RBD protein contains positive charges that helped against DNase digestion. I recommend the authors to check this paper "PMID: 28561045" for future vaccine fabrication. 10-hour stability looks limited to me.

Reviewer #3 (Remarks to the Author):

Review:

The manuscript by Remi Veneziano and co-workers describes a DNA origami based vaccine against the SARS-CoV 2 virus. The ongoing COVID19 pandemic has been devastating, and any research efforts aimed at developing preventive measures or treatments are highly appreciated. While numerous COVID vaccines have been developed, only a handful of them have been approved so far. Nanotechnology based vaccines have demonstrated great potential and its evident that these divergent set of platform technology would continue to be the frontrunners for any future vaccine development efforts.

While this manuscript is primarily aimed at showing the efficacy of yet another vaccine platform, additional discussions and even data (if available) will help us understand and appreciate this vaccine candidate better and compare it to other candidate vaccines. Please see the following comments:

Vaccine design: The authors have used DNA origami-based pentagonal bipyramid nanostructures as scaffolds to develop multivalent RBD vaccines, and have previously demonstrated that such scaffolds can be efficient vaccine platforms. The most exciting part of this vaccine design is the spatial control of antigen display that can be achieved. However, it would be informative to discuss why this particular geometry was selected from all other possible structures that they can generate. Was it determined by previous studies depicting enhanced lymphatic trafficking or APC uptake and activation of this structure over others? Or was it based on the ease of antigenic display? What was the rationale behind the asymmetric design of the nano-vaccine, with the adjuvants on one face and antigens on the other? Additional data or reference to previous work indicating the advantage of using this particular design would be useful and is recommended.

Stability of vaccine: FRET based studies have been performed to show the stability of the nano platform in biological media (serum). It is evident that the PB nanostructure is partially degraded over a 10h time, whereas PB-PG-RBD is slightly more stable, whereas PB-CpG is very stable. How about the complete vaccine RBD-PB-CpG? Is this vaccine which has been finally evaluated for efficacy, why not include the stability data on this? Also, some more discussion on the importance of stability or the importance of instability is warranted. The PB-vaccine is supposed to be taken up by APCs - is it important that the vaccine is easily processed, and the antigen/ epitopes presented? What is known about the eventual fate of the DNA origami based vaccine platform? If studied earlier, please refer to those discussions.

Immunogenicity and efficacy: The primary goal of this vaccine is to generate neutralizing antibodies. But, we now know that the longevity of this response is a critical issue in determining the effectiveness of any COVID vaccine. Therefore, one would expect to see results on long-term monitoring of the antibody titers, and not just for 3-weeks from the booster dose. The same could be said of the efficacy studies in ACE2 mice challenged with the virus, but it is understood that these studies are performed in BCL3 facilities and access/ resources limitations would only make short term studies feasible. However, the immunogenicity data in Balb/c mice could have been

studied longer, and it would have been nice to see data supporting the Th1 bias (antibody isotyping, if not ELISPOT from splenocytes) as claimed by the authors. Also, for both the immunogenicity and efficacy studies, RBD + CpG should have been another control to show the importance of PB nanostructure. Again, while complex and large studies in ACE2 mice are not feasible everywhere, at least immunogenicity in BALB/c mice should have included all possible controls.

Additionally, it would be nice to include some sentences on the feasibility of such a vaccine in context of manufacturing cost and scale up production. We already know that vaccine distribution is skewed in favor of rich nations and vaccine cost and availability is a critical bottleneck. Some statement addressing those issues would be appreciated without taking away anything from the potency of the candidate.

At this point, I would recommend revisions to the manuscript before publication. If the authors can add minor data, that would be very useful. If not, they should at least add more discussions regarding some of the points raised above.

Response to Reviewer #1.

The manuscript by Esra Oktay et al. describes a novel SARS-CoV2 vaccine formulation that utilizes protein antigen (RBD of SARSCoV2 virus), an adjuvant (CpG oligonucleotide), and a nanocarrier (DNA-origami nanoparticle). The authors present a physicochemical characterization of the DNA origami-based vaccines and efficacy studies in a mouse model.

Before publication, I suggest the authors consider several additional controls and questions summarized below:

We thank the reviewer for their thorough review and for their insightful comments. We have followed their suggestions in order to improve the quality of our manuscript. Our answers to the different points raised by the reviewer are presented below and all the changes made in our manuscript have been highlighted using a blue font for the reviewer's convenience.

1. The efficacy studies presented in Figures 3 and 4 would benefit from adding three controls - RBD-CpG conjugate, unconjugated RBD+CpG mixture, and RBD+PB+CpG unconjugated mixture of three components. These controls would help answer two very important questions about the need for a) the PB-component and b) conjugation of individual components for vaccine efficacy.

We thank the reviewer for the great suggestion regarding the missing control groups that would certainly strengthen the quality of our efficacy study by testing the effect of each component of our vaccine delivered in an unconjugated way. We have performed additional *in vivo* experiments using the following control groups including the suggestions from the reviewer and other controls that we deemed to be critical: *i.* A 1 µg dose of 3-mer RBD; *ii.* A 1 µg dose of 3-mer RBD + CpG (unconjugated); *iii.* A 1 µg dose of 3-mer RBD + DNA pentagonal bipyramid (unconjugated); *iv.* A 1 µg dose of 3-mer RBD + DNA Pentagonal bipyramid + CpG (unconjugated); *v.* A 1 µg dose of 3-mer RBD + Alhydrogel® as adjuvant added in the formulation (instead of CpG). We have added the information regarding the preparation of these control samples in a new Method section named '*Preparation of control groups*' and added some relevant information in the existing Method section called '*Animal immunization and viral challenge*'. The new results have been added in Figure 4 and 5 and in two different Results sections. In the Results section '*Immunizing BALB/c mice with DNA origami nanovaccines elicits high antibody response*', we have added the following text (page 15):

We also prepared five different control groups: 3-mer RBD (1 µg), 3-mer RBD (1 µg) + PB (unconjugated), 3-mer RBD (1 µg) + CpG (unconjugated), 3-mer RBD (1 µg) + PB + CpG (unconjugated), 3-mer RBD (1 µg) + Alhydrogel® to assess the effect of each component individually (Fig. 4).

In the Results section ‘*DNA origami nanovaccines protect against aerosol challenge with live SARS-CoV-2*’ section, we have added the following text (page 17):

The animals injected with the unconjugated samples showed a decrease in the survival rate after the viral challenge. For instance, we observed a mortality of 100% for the animals injected with 3-mer RBD (1 µg) + PB (unconjugated); 80% mortality for those injected with 3-mer RBD (1 µg) + CpG (unconjugated); 60% mortality with those injected with RBD (1 µg) or RBD (1 µg) + PB + CpG (unconjugated). Interestingly the animals injected with RBD (1 µg) + Alhydrogel® showed the lower mortality rate of all the control samples tested with only 40% mortality.

Our new results from the immunization and challenge experiments, suggest that the presence of adjuvants (either the CpG 1018 or the Alhydrogel®) is critical in increasing the immune response against RBD antigens (in trimeric form) even when unconjugated and free in solution. These results also confirm the importance of presenting the antigens in a multivalent way on the DNA nanoparticles to induce the strongest response (immunization and protection) for low quantity of antigen delivered (1 µg). Interestingly, our trimer RBD formulated with Alhydrogel® is able to induce 60% protection in the viral challenge experiments, which is similar to the protection observed with 5 µg of RBD trimer presented on the DNA nanoparticle without adjuvants.

2. Another important control is RBD+clinical grade adjuvant, for example, Alum

We thank the reviewer for this suggestion. We have included the control sample RBD (3-mer) + Alum (Alhydrogel®) in our control group as presented in the previous point. Relevant information regarding its preparation and the results from animal studies were included in Figure 4 and 5 and in the Results and Methods sections. The results were discussed in the previous reviewer point.

3. How does the efficacy of DNA-origami-based formulation compare to that of LNP-mRNA formulations that are already in clinical use?

We thank the reviewer for asking this question regarding the comparison with mRNA vaccines. We have not been able to perform a direct *in vivo* comparison our DNA-NP-based vaccine with the current LNP-based mRNA vaccines because we did not have access to any of these vaccines. However, it is important to note that mRNA vaccines are ‘nucleic acid-based’ vaccine, thus, they use a different mechanism for delivery of the antigens that first require delivery of intact mRNA and then its translation into proteins after release into the cytosol and prior to be presented by the antigen presenting cells. Moreover, the amount of mRNA vaccine administered is usually higher than 10 µg of mRNA and the estimated amount of protein translated from one mRNA molecule can vary significantly. For instance, a study performing head-to-head comparison between mRNA vaccine and subunit vaccine showed that the 0.3 µg injected mRNA resulted in 8 µg/ml RBD found in cell lysis supernatant, which indicates that the amount of protein translated is different than injected mRNA as expected for this type of system.¹ Because mRNA delivery is highly dependent on the type of lipid particle used and the size of the mRNA, it is difficult to determine how many

mRNA are encapsulated per LNPs and how many are delivered into the cells. Therefore, it is impossible to know the actual amount of protein delivered to each cell with our strategy and compare the neutralization efficacy, duration or the level of antibody responses between mRNA vaccine and our NP vaccine. With our new vaccine particles, we shown that 100% protection is reached in a hACE2 mouse model for as low as 1 µg of antigens in presence of CpG adjuvants. Interestingly, our results on viral neutralization and serum antibody levels (i.e., IgG) appear to be relatively similar with the results of some studies on mRNA vaccines.^{2,3} Regarding the immunization assay our strategy is showing a high level of neutralization right after the second dose (21 days) and the neutralization efficiency, while decreasing a little after two months (60 days) but remaining higher than the control samples tested. We have added some details regarding the mRNA vaccines and some comparison points with nanoparticle vaccines in different main text sections including the introduction and the results.

Ref. 1 Wu, Y., Zhang, H., Meng, L., Li, F. & Yu, C. Comparison of Immune Responses Elicited by SARS-CoV-2 mRNA and Recombinant Protein Vaccine Candidates. *Front Immunol* **13**, 906457 (2022).

Ref. 2 Golshani, M. *et al.* SARS-CoV-2 Specific Humoral Immune Responses after BNT162b2 Vaccination in Hospital Healthcare Workers. *Vaccines* **10**, 2038 (2022).

Ref. 3 Liu, J. *et al.* Correlation of vaccine-elicited antibody levels and neutralizing activities against SARS-CoV-2 and its variants. *Clinical and Translational Medicine* **11**, e644 (2021).

4. How sustainable is the immune response? How long does the protection last? The main disadvantage of LNP-mRNA vaccines is their short-term action which requires a boost after five months. Does the PB-RBD-CpG vaccine overcome this limitation?

We appreciate the reviewer questions. Duration of the response is indeed a critical parameter of the vaccines developed against SARS-CoV-2 (and any other pathogens) and a serious limitation of some of the mRNA vaccines available. To better characterize the immune response triggered by our new vaccine strategy and start answering this question, we have performed another set of *in vivo* immunization experiments to determine the level of immunization two months after the injection of the second dose. Due to financial and time constraints as well as the inherent complexity of BSL 3 work, we have only been able to conduct a two-month immunization study.

Data collected from the serum sample of the immunized mice with 1 µg 3-mer RBD-PB-CpG vaccine after two months showed that viral neutralization activity of antibodies is rather effective with a 50% viral inhibition with up to 1:1024 serum dilution. This result is higher than the immunization obtained with our control groups and close to the immunization levels obtained just after the second doses with the sample 1 µg 3-mer RBD-PB-CpG and 5 µg 3-mer RBD-PB. In the

Results section '*DNA origami nanovaccines protect against aerosol challenge with live SARS-CoV-2*', the following information was integrated into the text at page 19:

We calculated the titers for each type of antibody via ELISA assay (Supplementary Fig. 17) and evaluate the durability of the antibody responses in an extended period (2 months after the boost dose) using blood serum of animals immunized with 1 μ g RBD-PB-CpG vaccine construct. Strikingly, virus neutralization was observed up to a serum dilution of 1:1024, very close to the initial results obtained 15 days after the second injection (Fig. 6).

5. Does this formulation induce RBD-specific antibodies (IgM, IgG, IgA), and at what titer? What is the specificity of these antibodies? Does it induce RBD-specific cytotoxic T-cell response? Is there any cross-reactivity of the induced antigen-specific response to normal antigens? Does it induce RBD-specific cytotoxic T-cell response?

We thank the reviewer for this suggestion. We conducted additional *in vivo* studies with the new experiments done after 2 months. We performed ELISA assay with serum samples from the immunized mice with 1 μ g 3-mer RBD-PB-CpG vaccine which were collected after two-month and compared with the results of serum samples from mice injected with placebo. RBD-specific IgG concentration was found to be about 1,655.5 ng/ml (mean of N=4 mice), IgM concentration was about 28,031 ng/ml (mean of N=4 mice), and IgA was about 66 ng/ml (mean of N=4 mice) using RBD-specific ELISA kit from Creative Diagnostics®. Based on the information provided by Vendor, the assay specificity is high and no cross-reactivity was observed in experimental testing. The regarding information about the experimental study was added in the existing Methods section '*Animal immunization and viral challenge*' (page 31):

Enzyme-linked immune-sorbent assay (ELISA): ELISA kit for RBD-specific serum IgG, IgM, and IgA detection were used as described in the protocol provided by Creative Diagnostics®.

In the Results section '*DNA origami nanovaccines protect against aerosol challenge with live SARS-CoV-2*' along with the figure (Updated Figure 6, panel c), we also added following text (page 18)

Furthermore, to better understand the type of response triggered by our new vaccine, we assessed the RBD-specific antibodies (i.e., IgM, IgG, IgA) produced in the immunized mice. Indeed IgG and IgM are critical for immune protection against SARS-CoV-2 through humoral immunity and IgA are important for mucosal immunity, which is a key component of the response against SARS-CoV-2.⁶⁸ We calculated the titers for each type of antibody via ELISA assay (Supplementary Fig. 17) and evaluate the durability of the antibody responses in an extended period (2 months after the boost dose) using blood serum of animals immunized with 1 µg RBD-PB-CpG vaccine construct. Strikingly, virus neutralization was observed up to a serum dilution of 1:1024, very close to the initial results obtained 15 days after the second injection (Fig. 6). The concentration of RBD-specific IgG was found to be about 1,655.5 ng/ml, IgM was about 28,031 ng/ml, and IgA was about 66.0 ng/ml. The cross reactivity was not detected based on the tests applied by the manufacturer.

Unfortunately, due to the constraint of the BSL 3 work we were not able to examine cytotoxic T-cell response, which should be part of the next study if funding becomes available.

6. What is the role of the PB component? Does it have an adjuvant activity, deliver CpG and RBD to APCs, or both? How does the PB deliver antigen and adjuvant to APCs?

We apologize for not being clearer on the role of the PB component. We did not plan to use the PB scaffold as an adjuvant and recent work published have shown that the DNA origami nanoparticles have neglectable immunogenicity *in vivo*.^{4,5} This information from literature combined with our immunization and protection experiments using the control groups suggest that the PB component do not have any specific adjuvant effect. Indeed, when we compared the immunization and protection responses upon delivery of unconjugated mix of vaccine components, the contribution of PB delivered along with RBD seemed to be non-significant, which indicated there was not any significant stimulation by PB itself. Similarly, in comparison to another adjuvant ('Alhydrogel®'), PB does not increase the response.

To better distinguish the differences between the effect of PB and adjuvants (CpG and Alhydrogel®), the plot of the survival rate in the updated Figure 5 (panel b; upper plot) was described in the existing Results section '*DNA origami nanovaccines protect against aerosol challenge with live SARS-CoV-2*' at page 18.

We were not able to perform *in vitro* or *in vivo* tracking of our nanoparticle-based vaccine system. Earlier studies indicated that the composition, geometry, shape, or size vary the uptake time or mechanism of nanoparticles.^{6,7} However, the *in vivo* results of our nanoparticle vaccine suggested that the time for the uptake by antigen presenting cells might be enough for our nanoparticle to interact antigen presenting cells while it is still intact.

Ref. 4 Lucas, C. R. et al. DNA Origami Nanostructures Elicit Dose-Dependent Immunogenicity and Are Nontoxic up to High Doses In Vivo. *Small* **18**, 2108063 (2022).

Ref. 5 Du, R. R. et al. Innate Immune Stimulation Using 3D Wireframe DNA Origami. *ACS Nano* *acsnano.2c06275* (2022) doi:10.1021/acsnano.2c06275.

Ref. 6 Liu, X. et al. A DNA Nanostructure Platform for Directed Assembly of Synthetic Vaccines. *Nano Lett.* **12**, 4254–4259 (2012).

Ref. 7 Irvine, D. J., Aung, A. & Silva, M. Controlling timing and location in vaccines. *Advanced Drug Delivery Reviews* **158**, 91–115 (2020).

7. Is the PB-component immunogenic, i.e., does it induce DNA-origami specific antibody response? Can DNA origami break tolerance and induce DNA-specific antibodies? Answering these questions would help to eliminate autoimmunity concerns.

There are some earlier studies showing that the DNA origami nanoparticles are either non-immunogenic or negligibly immunogenic, which refer to that these nanoparticles are non-toxic or do not cause cytokine storm. Seemingly, the shape of the nanoparticles is one of key players in varying responses of immune cells.⁴ Despite the stimulation of immune responses by different shape of nanoparticles is yet to be found out, a study investigating wireframe PB NP (82 edge length) showed that this compact structure has activated TLR9 activation and interferon gamma (IFN λ) production in a negligible extent.⁵ For this reason, we have not specifically study the immunogenicity of DNA nanoparticles to evaluate the DNA-origami specific immune response.^{4,5,8,9} The further information was covered in the previous response.

A small paragraph including some of the main references listed here and discussing the immunogenic potential of the PB was included in the manuscript as follows (page 5):

In addition, recent studies have shown that DNA origami and particularly the pentagonal bipyramid have negligible immunogenicity, especially in regards to the stimulation of the TLR9 pathway, which is important to determine the specificity of the vaccine response induced by the antigens.^{41,42}

Ref. 4 Lucas, C. R. et al. DNA Origami Nanostructures Elicit Dose-Dependent Immunogenicity and Are Nontoxic up to High Doses In Vivo. *Small* **18**, 2108063 (2022).

Ref. 5 Du, R. R. et al. Innate Immune Stimulation Using 3D Wireframe DNA Origami. *ACS Nano* *acsnano.2c06275* (2022) doi:10.1021/acsnano.2c06275.

Ref. 8 Schüller, V. J. et al. Cellular Immunostimulation by CpG-Sequence-Coated DNA Origami Structures. *ACS Nano* **5**, 9696–9702 (2011).

Ref. 9 Surana, S., Shenoy, A. R. & Krishnan, Y. Designing DNA nanodevices for compatibility with the immune system of higher organisms. *Nat Nanotechnol* **10**, 741–747 (2015).

Response to Reviewer #2:

This manuscript developed a DNA origami nanoparticle-based vaccine for Sars-Cov-2 by co-

delivering RBD antigen and CpG adjuvant. The authors described the vaccine fabrication methods thoroughly and proved the vaccine efficacy in health mice immunization and mouse virus challenge study.

The overall impression of this manuscript: 1). It's well written in detail and easy to understand; 2). It provided mouse studies to evaluate the vaccine efficacy. 3). However, there are several major concerns to be addressed before the authors can make the conclusion that the nanoparticle really helps.

We would like to thank the Reviewer for these encouraging comments about our manuscript and our strategy and apologize for the lack of clarity in some part of the manuscript. We have tried to answer all comments and questions in the following paragraph and have edited our manuscript and SI document accordingly. We hope that these changes will satisfy the reviewer and help support our conclusions.

Majors:

1. Antigen and adjuvant are essential to stimulate strong immune responses. The authors also cited the FDA-approved vaccine HBsAg-CpG in 2018. However, the researchers in this work didn't include proper control in the animal study to compare with RBD and CpG given freely (we call bolus vaccine control). That is the key to show the advantage of the DNA origami nanoparticle. Otherwise, why would people use complicated and expensive DNA origami? Both figure 3 and figure 4 should include this control.

We appreciate the reviewer's comment. We agree that the including control samples will help better understand and validate the conclusions of our study. Thus, we have designed new *in vivo* studies to include the following control groups: *i.* 1 µg dose of 3-mer RBD; *ii.* Unconjugated mix of CpG and 1 µg dose of 3-mer RBD; *iii.* Unconjugated mix of DNA pentagonal bipyramid and 1 µg dose of 3-mer RBD; *iv.* Unconjugated mix of DNA Pentagonal bipyramid, CpG and 1 µg dose of 3-mer RBD; and *v.* Alhydrogel® and 1 µg dose of 3-mer RBD. We have added the information regarding the preparation of these control samples in a new Method section called '**Preparation of control groups**' and relevant information were included in the existing Method section called '**Animal immunization and viral challenge**'.

The new results have been inserted in Figures 4 and 5 (updated old Figures 3 and 4) and in two different Results section. In the Results section '**Immunizing BALB/c mice with DNA origami nanovaccines elicits high antibody response**', we have added the following text (page 15):

We also prepared five different control groups: 3-mer RBD (1 µg), 3-mer RBD (1 µg) + PB (unconjugated), 3-mer RBD (1 µg) + CpG (unconjugated), 3-mer RBD (1 µg) + PB + CpG (unconjugated), 3-mer RBD (1 µg) + Alhydrogel® to assess the effect of each component individually (Fig. 4).

In the Results section ‘*DNA origami nanovaccines protect against aerosol challenge with live SARS-CoV-2*’ section, we have added the following text (page 17).

The animals injected with the unconjugated samples showed a decrease in the survival rate after the viral challenge. For instance, we observed a mortality of 100% for the animals injected with 3-mer RBD (1 µg) + PB (unconjugated); 80% mortality for those injected with 3-mer RBD (1 µg) + CpG (unconjugated); 60% mortality with those injected with RBD (1 µg) or RBD (1 µg) + PB + CpG (unconjugated). Interestingly the animals injected with RBD (1 µg) + Alhydrogel® showed the lower mortality rate of all the control samples tested with only 40% mortality.

Immunization and challenge studies indicated that the co-delivering adjuvants (CpG ODN 1018 or Alhydrogel®) with antigen either freely or conjugated to NP help augment the immune response. Similarly, formulating RBD with Alhydrogel® (RBD adsorbed on Alhydrogel in a 1:20 mass ratio) showed 60% protection in the viral challenge experiments, which is similar to the protection observed with 5 µg of (3-mer) RBD-PB DNA-NP vaccine without adjuvants. Despite, the most protective response was achieved when presenting multiple CpG on DNA-NP along with multiple copy of antigen, which helped reduce the quantity of antigen used (1 µg).

2. TEM images should be included to show the robust vaccine fabrication. I don’t see any TEM results in this manuscript, which is not acceptable for DNA origami studies. In figure 2a, when 5 times more of RBD was used to incubate with PB, the band shift looks to me like aggregations. This can be caused by protein fabrication with DNA structures. TEM images should be able to tell if higher-order aggregation was formed.

Given the nature of the DNA-NPs used and their size and shape, we decided to use AFM as it appear more adapted for imaging these almost flat structures as previously shown in different papers using these structures.¹⁰ We understand that electrophoresis is not the most adapted method to characterize and quantify the conjugation of protein on the DNA-NPs and serve as a confirmation of a change of molecular weight of the nanoparticles. Therefore, we have also performed many other types of experiments to better characterize conjugation efficiency. For instance, our DLS measurements indicated that bare PB nanoparticles, PB nanoparticles with protein G and PB nanoparticles with RBD are highly monodisperse and show slightly increased in their hydrodynamic diameter with stepwise protein functionalization. These results also show that our nanoparticles are not aggregating after conjugation. Moreover, AFM, fluorescence emission-based stoichiometry, and mass spectroscopy results also contributed to validating the findings of DLS. Details of these techniques and the results were introduced in the Methods (starting at page 24) and Results section (starting at page 8). The new Figure 2 covering the results of all methods was included in the manuscript.

Ref. 10 Veneziano, R. *et al.* Role of nanoscale antigen organization on B-cell activation probed using DNA origami. *Nat. Nanotechnol.* **15**, 716–723 (2020).

3. I don't see a vaccine purification method mentioned in the manuscript. I also don't see the determination of component conjugation efficiency (e.g. CpG and RBD). It's not like 10 overhangs included, then 10 molecules will be conjugated. This can be determined by DNase digestion and running denature page for CpG or SDS page for protein. The purification and conjugation efficiency should be essential to vaccine fabrication.

We apologize for missing these critical details in our manuscript regarding the vaccine purification and characterization. We have performed purification using size-based spin-column filtration to remove the excess of 3-mer RBD or CpGs. The Methods section '*Preparation of the DNA origami nanoparticle nanovaccine*' proceeding with subtitle '*Functionalization of the PB nanoparticle with the RBD trimers and the CpG strands*' at page 24 provided the information regarding the process of purification, as follows:

Reaction product was filtered at least 3 times from excess amount of CpG and PG-RBD protein complex using centrifugal filter (300 kDa MWCO).

In addition, to characterize our nanovaccine in a qualitative and quantitative way, we have developed multiple strategies that allowed us to determine the stoichiometry of PG and RBD and the ratio of RBD-to-PG. To determine the stoichiometry of proteins (PG and RBD), we conducted two different fluorescence-based assays. The percentage of DNA-NP coverage by Protein G was quantified (in triplicate) based on the measurement of intrinsic tryptophan fluorescence intensity as mentioned in an earlier study¹¹. According to the DNA-NP concentration measured via Nanodrop, we estimated the theoretical Protein G concentration by multiplying the DNA-NP concentration by 10 (10 sites). Using a standard curve of tryptophan and a standard sample of free protein G, we have determined the concentration of protein G attached on the DNA-NP. Specifically, we created a standard curve based on different concentrations of free tryptophan (0 to 20 μ M) and calculated the free PG-PNA and PG-PNA on DNA-NP concentration. The coverage of DNA-NP with PG-PNA was found to be about 101% \pm 11.4 (mean \pm std). The information regarding the entire experimental procedure was detailed in the Methods section under a subsection titled as '*Measuring intrinsic tryptophan fluorescence intensity of PG on NP*' at page 25-26, where the following text was included.

Utilizing the optical properties of tryptophan due to the aromatic ring, we established a standard curve based on the fluorescence emission of varying concentrations of tryptophan (concentration range between 0 to 20 μ M). Particularly, to distinguish the emission of tyrosine from tryptophan, excitation wavelength was set to 295 nm. Emission wavelength was set from 325 nm to 500 nm. DNA-NP folded by the staples with 10 overhangs reacted with 2-fold molar excess of PG-PNA and purified from excess PG-PNA. Based on NP concentration, we estimated the protein concentration on the NP. By using tryptophan standard curve, free PG-PNA and NP with PG-PNA which were assumed having same concentration of PG were calculated.

Following text regarding the results were included in the updated Results section '***Determination of the stoichiometry of PG and RBD on DNA-NP***' at page 9 and presented in Figure 2 (panel c);

To quantitatively determine the level of conjugation, we used fluorescence measurements. We first used tryptophan fluorescence assays to determine the level of PG modification on DNA-NP displaying 10 overhangs (Supplementary Fig. 11). Based on the fluorescence measurements, the percentage of coverage of the DNA-NP with PG was found to be close to 100% (i.e., $101.2 \pm 11.4\%$, mean \pm SD of N=3 distinct replicates) (Fig. 2c).

The number of RBD on DNA-NP was determined based on Cy5 intensity. Cy5 was initially conjugated to RBD via NHS-amine coupling and then RBD-Cy5 was attached to DNA-NP through conjugation to PG. A Free Cy5 standard curve with Cy5 in a concentration range of 0-10 μ M was used to establish a standard curve and based on this curve we calculated the coverage of DNA-NP by RBD-Cy5 which was found to about 77%. The detailed information as follows was integrated in the Methods section '***Assessment of RBD stoichiometry on NP via measuring Cy5 emission***' (page 26):

The labeling reaction of RBD with Cy5-NHS was performed overnight at 4°C. Cy5-NHS was mixed with RBD (at a ratio of 1:10) in sodium bicarbonate (NaHCO₃) pH 8.4. Next day, the excess Cy5-NHS was removed from the solution via spin-filtration using Amicon Filter. In the second step, the Cy5-NHS-labeled RBD was attached to the NP through incubating at 37°C. The standard curve (concentration range between 0 to 10 μ M) was established according to the emission from Cy5 dye on RBD, in which the excitation was set at 590 nm and emission spectrum range was arranged between 590 nm and 640 nm (Biotek Cytation 5 Cell Imaging Multimode Reader).

The relevant result were added in the Results section '***Determination of the stoichiometry of PG and RBD on DNA-NP***' (page 9-10) and the results regarding the coverage of DNA-NP by RBD were shown as a bar graph in Figure 2 (page 11):

Further, by using a fluorescently labelled version of the RBD protein (cy5 labelled), we were able to measure the percentage of the coverage of the DNA-NP presenting 10 copies of PG. Our results show that about 77% (Fig. 2c) of the available sites (30 sites) on the PG were occupied by the RBD proteins (i.e., ~23 RBD domains for 10 overhangs) (Supplementary Fig. 12).

We also included the information regarding the relative quantification of RBD:PG ratio on DNA-NP via mass spectrometry. The experimental procedure was detailed in the Methods section ‘*Mass spectrometry for quantifying PG to RBD ratio on NP*’ (page 26) and ‘*Liquid Chromatography with tandem mass spectrometry (LC-MS/MS) method*’ (page 27) and the molar ratio of RBD:PG was presented in the Results section ‘*Measuring the RBD:PG ratio on DNA-NP via Mass Spectrometry*’ (text from page 10 provided below). Based on the mass spectra (provided in Supplementary Fig. 13), the molar ratio between RBD and PG was found to be 2.82:1 (RBD:PG), wherein theoretical ratio is supposed to be 3:1. This result reinforces the data obtained from the stoichiometry analysis for DNA-NP coverage by each individual protein via fluorescence emission spectrum.

Our results indicate that the molar ratio between RBD and PG was determined to be 2.82:1 (RBD:PG) for a theoretical ratio of 3:1 (Supplementary Fig. 14), which confirmed the efficacy of our conjugation strategy.

Ref. 11 Wiśniewski, J. R. & Gaugaz, F. Z. Fast and Sensitive Total Protein and Peptide Assays for Proteomic Analysis. *Anal. Chem.* **87**, 4110–4116 (2015).

Minors:

1. Line 120, the second sentence of the Fig. 2a legend is not clear to me.

Thank you for pointing out. To clarify, we first reacted PG with RBD separately and afterwards we incubated PG-RBD protein mixture with DNA-NP. PG-RBD concentration was adjusted for DNA-NP incubation as 2:1 molar ratio (PG:DNA-NP).

2. Line 216, the authors mentioned there are no adverse events. But I don't see any supporting data for this.

Thank you for pointing out. The adverse events mentioned here were all based on observations on mice regarding the changes in their physical appearance, mobility, attitude, body features during the period after administration of vaccine and all changes were recorded based on the scores given according to ‘Animal Study Clinical Monitoring Chart’ scores. We included informatory tables in Supplementary Information (Supplementary Table 9 and Supplementary Table 10).

3. Line 170, it's interesting the authors mentioned: “coating DNA-NPs with protein and modified nucleic acids can participate in increasing the stability against nucleases without the need for complex chemical modifications as seen in extant literature”. Maybe the RBD protein contains positive charges that helped against DNase digestion. I recommend the authors to check this paper “PMID: 28561045” for future vaccine fabrication. 10-hour stability looks limited to me.

Thanks for this suggestion. It would be indeed very useful to clarify what are the possible reasons

for the increased stability of nanoparticles when modified with proteins and/or CpGs. This specific problem was studied in a very recent published work where the researchers used CpG ODNs on DNA-NPs to investigate the activation level of TLR9 signaling pathway. Specifically, they evaluated the level of CpG degradation by nucleases in physiological serum. They found that CpGs on DNA NP as well as free phosphorothioate-modified CpGs are stable against nuclease degradation as previously shown in literature. This might be one way of explaining how the presence of phosphorothioate CpGs on DNA-NP participate in reinforcing the stability of NPs by preventing degradation from the 3' end or nicks of the DNA-NPs.⁵ Coating with positively charged molecules can also enhance the stability of nanoparticles. As an example, oligolysine-coated DNA nanoparticles have been found to be more stable in serum for more than 24 hours against negatively charged DNase I. This was particularly visible when oligolysine molecules were presented with a linker (i.e., PEG), which also prevents aggregation of DNA nanoparticles due to the interference of positively charged molecules with negatively charged DNA phosphate groups.¹² Coating DNA nanoparticle with any molecule serve for protection against protein corona effect or nuclease digestion by preventing access of the nuclease to the DNA backbone. In addition, the charge of molecules is critical to avoid aggregation of DNA nanoparticles due to charged-based repulsion of strands.¹³ Particularly, in our strategy we have a neutrally charged PNA linker between Protein G and DNA NP that provides indirect contact between proteins and DNA NP. To investigate the stability of nanoparticles for a longer time, we prolonged the incubation in serum up to 24 hours. Three different nanoparticle vaccine samples (i.e., PB-CpG, PB-RBD, and RBD-PB-CpG) were monitored in a time-based study and CpG, RBD or RBD/CpG-coated nanoparticles have been found to be 40% more stable than the unmodified DNA nanoparticles.

Ref. 5 Du, R. R. *et al.* Innate Immune Stimulation Using 3D Wireframe DNA Origami. *ACS Nano* acsnano.2c06275 (2022) doi:10.1021/acsnano.2c06275.

Ref. 12 Ponnuswamy, N. *et al.* Oligolysine-based coating protects DNA nanostructures from low-salt denaturation and nuclease degradation. *Nat Commun* **8**, 15654 (2017).

Ref. 13 Auvinen, H. *et al.* Protein Coating of DNA Nanostructures for Enhanced Stability and Immunocompatibility. *Advanced Healthcare Materials* **6**, 1700692 (2017).

Response to Reviewer #3:

Review:

The manuscript by Remi Veneziano and co-workers describes a DNA origami based vaccine against the SARS-CoV 2 virus. The ongoing COVID19 pandemic has been devastating, and any research efforts aimed at developing preventive measures or treatments are highly appreciated. While numerous COVID vaccines have been developed, only a handful of them have been approved so far. Nanotechnology based vaccines have demonstrated great potential and its evident that these divergent set of platform technology would continue to be the frontrunners for any future vaccine development efforts.

While this manuscript is primarily aimed at showing the efficacy of yet another vaccine platform, additional discussions and even data (if available) will help us understand and appreciate this vaccine candidate better and compare it to other candidate vaccines. Please see the following comments:

We would like to thank the Reviewer for their kind comments and their suggestions to improve the quality of our manuscript. We have answered all the questions asked and we hope that the quality of our updated manuscript will match the requirements for publication in *Communications Biology*.

Vaccine design: The authors have used DNA origami-based pentagonal bipyramid nanostructures as scaffolds to develop multivalent RBD vaccines, and have previously demonstrated that such scaffolds can be efficient vaccine platforms. The most exciting part of this vaccine design is the spatial control of antigen display that can be achieved. However, it would be informative to discuss why this particular geometry was selected from all other possible structures that they can generate. Was it determined by previous studies depicting enhanced lymphatic trafficking or APC uptake and activation of this structure over others? Or was it based on the ease of antigenic display? What was the rationale behind the asymmetric design of the nano-vaccine, with the adjuvants on one face and antigens on the other? Additional data or reference to previous work indicating the advantage of using this particular design would be useful and is recommended.

We appreciate the reviewer comments. We agree that the nanoparticles shape and size is critical for interactions and internalization by cells.¹⁴⁻¹⁶ Our choice was based on multiple parameters and also on our previous study with HIV antigens published in 2020 in *Nature Nanotechnology*. Primarily, we chose the pentagonal bipyramid shape which offers optimal surface area with max. total 30 overhangs on surface and edges together without the necessity for using larger constructs.¹⁰ Also, the particular features of PB was included in the text of updated manuscript as follows;

Particularly, the almost flat surface of the PB makes it similar to oblate ellipsoidal nanoparticles that are known to be preferentially uptaken by immune cells due to the larger surface area of interaction and the receptor diffusion kinetics, which facilitate membrane wrapping and internalization based on simulations and *in vitro* experiments.^{37,38,40} Moreover, our PB DNA origami NP also provides two surfaces with the same geometries that allow multiplexed presentation of various biomolecules simultaneously (Supplementary Fig. 1).

Moreover, by having two faces, PB structure provides simultaneous presentation of distinct molecules for enhanced immune response. By decorating one face of PB NP with 3-mer RBD, we intended to stimulate immune cells to develop counter response against virus RBD antigen. Since CpG is a clinically approved adjuvant and has been also used in some of the SARS-CoV-2 vaccine studies to enhance the immune response^{17–19}, we also preferentially formulated our vaccine constructs using CpGs on the opposite side of RBD. We assumed that once the nanoparticles are uptaken into the endosome upon recognition of antigens on one face of NP, CpG on the other face of NP might be delivered for interacting with the intracellular TLR9 receptors. In case of DNA-NP falling apart in the process of internalization through endosomal and later lysosomal degradation, CpG ODNs might still remain intact due to their fully phosphorothiated backbone, and it might activate internal TLR9 pathway. Indeed, a recent study showed that the DNA-NP functionalized with regular CpGs acted against nuclease degradation of CpGs and thereby proceeded for the activation of TLR9 pathway.⁵ Phosphorothioate-modified CpGs are even more stable on DNA-NP and have an important role in activation of endosomal TLR9 signaling pathway.

Ref. 5 Du, R. R. *et al.* Innate Immune Stimulation Using 3D Wireframe DNA Origami. *ACS Nano* [acsnano.2c06275](https://doi.org/10.1021/acsnano.2c06275) (2022) doi:10.1021/acsnano.2c06275.

Ref. 14 Bastings, M. M. C. *et al.* Modulation of the Cellular Uptake of DNA Origami through Control over Mass and Shape. *Nano Lett.* **18**, 3557–3564 (2018).

Ref. 15 Lacroix, A. & Sleiman, H. F. DNA Nanostructures: Current Challenges and Opportunities for Cellular Delivery. *ACS Nano* **15**, 3631–3645 (2021).

Ref. 16 Rajwar, A. *et al.* Geometry of a DNA Nanostructure Influences Its Endocytosis: Cellular Study on 2D, 3D, and *in Vivo* Systems. *ACS Nano* **16**, 10496–10508 (2022).

Ref. 17 Kuo, T.-Y. *et al.* Development of CpG-adjuvanted stable prefusion SARS-CoV-2 spike antigen as a subunit vaccine against COVID-19. *Sci Rep* **10**, 20085 (2020).

Ref. 18 Nanishi, E. *et al.* An aluminum hydroxide:CpG adjuvant enhances protection elicited by a SARS-CoV-2 receptor binding domain vaccine in aged mice. *Sci. Transl. Med.* **14**, eabj5305 (2022).

Ref. 19 Grigoryan, L. *et al.* Adjuvanting a subunit SARS-CoV-2 vaccine with clinically relevant adjuvants induces durable protection in mice. *npj Vaccines* **7**, 1–14 (2022).

Stability of vaccine: FRET based studies have been performed to show the stability of the nano platform in biological media (serum). It is evident that the PB nanostructure is partially degraded over a 10h time, whereas PB-PG-RBD is slightly more stable, whereas PB-CpG is very stable. How about the complete vaccine RBD-PB-CpG? Its this vaccine which has been finally evaluated for efficacy, why not include the stability data on this? Also, some more discussion on the importance of stability or the importance of instability is warranted. The PB-vaccine is supposed to be taken up by APCs - is it important that the vaccine is easily processed, and the antigen/ epitopes presented? What is known about the eventual fate of the DNA origami based vaccine platform? If studied earlier, please refer to those discussions.

We are sorry for the confusion and missing information. Earlier experiments on stability were performed over 10-hour for the NP vaccine constructs (RBD-PB-CpG and CpG-PB). We have prolonged the period for the assessment of stability up to 24-hour in 20% mouse serum along with the inclusion of RBD-PB NP construct. Stability is important parameter sought in establishing an ideal platform of nanoparticle-based vaccine delivery for maximum stimulation time for immune cell activation.²⁰ The protection of DNA origami NP, which function as a delivery vehicle of antigens and adjuvants immobilized on NP in rationally organized manner, is more critical to obtain effective response. Here, we have not studied the fate of the DNA-NPs once injected and internalized by immune cells. However based on the previous studies (including one from Dr. Veneziano on B cell activation with DNA-NPs presenting HIV antigens), DNA-NPs and proteins (including antigens) they carried could be internalized by different type of cells.^{10,21–23} Since yet there is not clear explanation regarding the way of uptake for particularly PB nanoparticles, we hypothesized that upon recognition of RBD by the host cell receptors on our PB nanoparticles, internalization would start through endocytic pathway. Once CpGs are localized inside the endosome, they would interact with TLR9 receptors which eventually induce the downstream signaling pathways for the ultimate response. As shown in previous studies, stable DNA nanoparticles which were achieved by coating with proteins were able to stimulate immune response. Therefore, the time for presentation of antigens to antigen presenting cells were assumed to be enough.⁶

Ref. 6 Liu, X. *et al.* A DNA Nanostructure Platform for Directed Assembly of Synthetic Vaccines. *Nano Lett.* **12**, 4254–4259 (2012).

Ref. 10 Veneziano, R. *et al.* Role of nanoscale antigen organization on B-cell activation probed using DNA origami. *Nat. Nanotechnol.* **15**, 716–723 (2020).

Ref. 20 Pati, R., Shevtsov, M. & Sonawane, A. Nanoparticle Vaccines Against Infectious Diseases. *Frontiers in Immunology* **9**, (2018).

Ref. 21 Vindigni, G. *et al.* Receptor-Mediated Entry of Pristine Octahedral DNA Nanocages in Mammalian Cells. *ACS Nano* **10**, 5971–5979 (2016).

Ref. 22 Wang, P. *et al.* Visualization of the Cellular Uptake and Trafficking of DNA Origami Nanostructures in Cancer Cells. *J. Am. Chem. Soc.* **140**, 2478–2484 (2018).

Ref. 23 Green, C. M., Mathur, D. & Medintz, I. L. Understanding the fate of DNA nanostructures inside the cell. *J. Mater. Chem. B* **8**, 6170–6178 (2020).

Immunogenicity and efficacy: The primary goal of this vaccine is to generate neutralizing antibodies. But, we now know that the longevity of this response is a critical issue in determining the effectiveness of any COVID vaccine. Therefore, one would expect to see results on long-term monitoring of the antibody titers, and not just for 3-weeks from the booster dose. The same could be said of the efficacy studies in ACE2 mice challenged with the virus, but it is understood that these studies are performed in BCL3 facilities and access/ resources limitations would only make short term studies feasible. However, the immunogenicity data in Balb/c mice could have been studied longer, and it would have been nice to see data supporting the Th1 bias (antibody isotyping, if not ELISPOT from splenocytes) as claimed by the authors. Also, for both the immunogenicity and efficacy studies, RBD + CpG should have been another control to show the importance of PB nanostructure. Again, while complex and large studies in ACE2 mice are not feasible everywhere, at least immunogenicity in BALB/c mice should have included all possible controls.

We thank the reviewer for pointing out the critical need for including control groups, which we also strongly agree. Based on all of these comments, first we included five different control samples: *i* free 3-mer RBD (1 µg); *ii*. 3-mer RBD (1 µg) + CpG (unconjugated); *iii*. 3-mer RBD (1 µg) + PB (unconjugated); *iv*. 3-mer RBD (1 µg) + PB + CpG (unconjugated); and *v*. 3-mer RBD (1 µg) + Alhydrogel®. The control samples were also injected as two doses (50 µl each dose) intramuscularly with a three-week interval between each dose injection. In the Methods section '**Preparation of control groups**', we included information about the procedure to prepare the control samples (page 24) and we also added follow-up information in the existing Method section called '**Animal immunization and viral challenge**' regarding the *in vivo* experimental procedures (page 29-30). Following texts from Methods section were provided as follows:

All control groups were composed of the unconjugated mix of separate components from the DNA-NP vaccine formulations wherein they were not incubated long time and at specific temperature. Control groups include the injection of 50 µl placebo (1X PBS), free 3-mer RBD (1 µg), 3-mer RBD (1 µg) + PB (unconjugated), 3-mer RBD (1 µg) + CpG ODN 1018 (unconjugated), 3-mer RBD (1 µg) + PB + CpG ODN 1018 (unconjugated), 3-mer RBD (1 µg) and Alhydrogel® per animals for the immunization and challenge part of the study as comparisons. As for RBD + wet gel alum (Alhydrogel®), 1 µg of RBD was mixed with Alhydrogel® in a 1.1:1 volume ratio (1:20 mass ratio) for 50µl injection per animal, at least 30 minutes before injection to promote sufficient adsorption of RBD on Alhydrogel®. Particularly, based on the previous studies referring the adsorption capacity of aluminum hydroxide-containing adjuvants on proteins^{70,71}, besides the information provided from vendor, led us to use this volume ratio between protein and adjuvant.

We prepared solution of 50 μ l of four different PB DNA-NP nanovaccine constructs: PB alone, RBD-PB (1 μ g and 5 μ g doses), RBD-PB-CpG (1 μ g dose) and five different control groups: free 3-mer RBD (1 μ g), 3-mer RBD (1 μ g) + CpG (unconjugated), 3-mer RBD (1 μ g) + PB (unconjugated), 3-mer RBD (1 μ g) + PB + CpG (unconjugated), 3-mer RBD (1 μ g) + Alhydrogel® that were administered to 6 to 8 weeks old female BALB/c mice via intramuscular (IM) injection in the right caudal thigh muscle. Two injections were done with three-week intervals.

The new results have been also added in Figure 4 and 5 and in two different Results sections. In the Results section '*Immunizing BALB/c mice with DNA origami nanovaccines elicits high antibody response*' and '*DNA origami nanovaccines protect against aerosol challenge with live SARS-CoV-2*' we have added the following texts (page 15 and page 17):

Our PRNT results demonstrate that viral inhibition by serially diluted serum antibodies from control groups were found not to be effective (Fig. 4b). Particularly, serum antibodies collected from the animals injected with 3-mer RBD (1 μ g) + PB + CpG (unconjugated) showed relatively high neutralization capacity followed by the even higher neutralization capacity of the antibodies from the 3-mer RBD (1 μ g) + Alhydrogel® injected animals in a serum dilution less than 1:1024. The neutralization capacity of antibodies produced by the animals injected with 3-mer RBD (1 μ g) alone, 3-mer RBD (1 μ g) with PB (unconjugated), or 3-mer RBD (1 μ g) with CpGs (unconjugated) were even lower with neutralization efficacy reduced drastically for serum dilutions superior to 1:32.

The animals injected with the unconjugated samples showed a decrease in the survival rate after the viral challenge. For instance, we observed a mortality of 100% for the animals injected with 3-mer RBD (1 μ g) + PB (unconjugated); 80% mortality for those injected with 3-mer RBD (1 μ g) + CpG (unconjugated); 60% mortality with those injected with RBD (1 μ g) or RBD (1 μ g) + PB + CpG (unconjugated). Interestingly the animals injected with RBD (1 μ g) + Alhydrogel® showed the lower mortality rate of all the control samples tested with only 40% mortality.

In a two months extension of our immunization study with injection of 1 μ g 3-mer RBD-PB-CpG DNA-NP and placebo (1X PBS) as a control, we analyzed the durability of antibody response by

checking the virus neutralization via plaque reduction neutralization assay and determined the level of RBD-specific serum antibodies via ELISA assay. The Methods section was updated with the information regarding the procedures under the existing methods '*Animal immunization and viral challenge*' and with new section for ELISA '*Enzyme-linked immune-sorbent assay (ELISA)*' (page 31-32). In the existing Results section '*DNA origami nanovaccines protect against aerosol challenge with live SARS-CoV-2*', the new information was included as follows (page 19):

Furthermore, to better understand the type of response triggered by our new vaccine, we assessed the RBD-specific antibodies (i.e., IgM, IgG, IgA) produced in the immunized mice. Indeed IgG and IgM are critical for immune protection against SARS-CoV-2 through humoral immunity and IgA are important for mucosal immunity, which is a key component of the response against SARS-CoV-2.⁶⁸ We calculated the titers for each type of antibody via ELISA assay (Supplementary Fig. 17) and evaluate the durability of the antibody responses in an extended period (2 months after the boost dose) using blood serum of animals immunized with 1 μ g RBD-PB-CpG vaccine construct. Strikingly, virus neutralization was observed up to a serum dilution of 1:1024, very close to the initial results obtained 15 days after the second injection (Fig. 6). The concentration of RBD-specific IgG was found to be about 1,655.5 ng/ml, IgM was about 28,031 ng/ml, and IgA was about 66.0 ng/ml. The cross reactivity was not detected based on the tests applied by the manufacturer.

Additionally, it would be nice to include some sentences on the feasibility of such a vaccine in context of manufacturing cost and scale up production. We already know that vaccine distribution is skewed in favor of rich nations and vaccine cost and availability is a critical bottleneck. Some statement addressing those issues would be appreciated without taking away anything from the potency of the candidate.

We appreciate the reviewer suggestion regarding scaling up our vaccine nanoparticle and understand that being able to scale up the production in a way that would help future global distribution of this new vaccine strategy for SARS-CoV-2 or other pathogens, is critical. While the DNA origami technology and the DNA origami-based vaccine strategy is a relatively recent method in comparison to other vaccine strategies, numerous groups, including our group are working toward scaling up the production of ssDNA scaffold and simplifying the assembly process to reduce the number of steps and cut the costs. We also think that because our strategy enhances the immunization and protection triggered by antigens even at low doses when presented in a rational way, which will contribute to reduction of the cost. In addition, our strategy relies on commercially available antigens and chemistry, which drastically reduce the cost in comparison with custom made nanoparticles and antigens. To discuss this point we have added the following paragraph in our main text (page 20).

In addition to paving the way toward using DNA-NPs as the potential next generation of nanoparticle-based vaccine, our strategy demonstrate that rational design of DNA-NP-based vaccine will be key to reduce the high cost associated with this new technology. Indeed, we have shown that properly organizing antigens and co-presenting adjuvants contribute to a significant decrease in the required quantity of antigen to induce strong immunization and protection in a mouse model. The controlled organization of the antigens and adjuvants simultaneously in the DNA-NP will greatly reduce the overall cost per dose of this novel DNA-NP-based vaccine strategy. However, it is important to note that new strategies for large-scale production of ssDNA scaffold are still required to reduce the cost of synthesis of these DNA-NPs. Additionally, updated methodologies to optimize particle purification in scaled up production will be beneficial to reduce the number of purification steps and increase the final production yield. Altogether, the results presented in this manuscript demonstrate the potential and the robustness of this novel vaccine strategy and pave the way toward rational design of vaccine nanoparticles that could be used to rapidly respond to future viral threats.

At this point, I would recommend revisions to the manuscript before publication. If the authors can add minor data, that would be very useful. If not, they should at least add more discussions regarding some of the points raised above.

We appreciate the reviewer comments and we hope our new results and discussion have improved our manuscript.

REVIEWERS' COMMENTS:

Reviewer #1 (Remarks to the Author):

The authors have conducted additional studies and addressed most of my comments; whenever they did not address some comments, they provided valid reasons, such as the unavailability of LNP-mRNA covid vaccines for research. The authors substantially edited the manuscript to provide clarification and explain the new, more comprehensive data sets. I recommend the manuscript for publication in its current form.

As a side note, it is important not to confuse immunogenicity, which refers to the generation of antigen-specific antibodies [humoral immunity] and antigen-specific T-cells [cellular immunity], with immunostimulation, which refers to cytokine secretion [cytokine storm if too high levels of cytokines are produced), complement activation, TLR stimulation. Many published papers are confusing and misleading because they use the term "immunogenicity" when describing cytokine responses, i.e. "immunostimulation."

Reviewer #2 (Remarks to the Author):

The authors did a nice job addressing my concerns and some of the other reviewers' concerns.

I don't see the CpG conjugation efficiency validation. Based on our study, it's not always 100% conjugated.

I also have one question for figure 6c. What are the comparing groups there? Shouldn't it be the different vaccine formulas you are comparing? I don't see the need to compare the different Igs.

Another minor: in the method section for control groups, could you also add the amount of CpG and nanoparticles beside the amount of antigen?

Response to Reviewers (Manuscript # COMMSBIO-22-0381), 2nd revision

Response to Reviewer #1.

The authors have conducted additional studies and addressed most of my comments; whenever they did not address some comments, they provided valid reasons, such as the unavailability of LNP-mRNA covid vaccines for research. The authors substantially edited the manuscript to provide clarification and explain the new, more comprehensive data sets. I recommend the manuscript for publication in its current form.

We thank the reviewer for their thorough review of our revised manuscript and for their positive comments.

As a side note, it is important not to confuse immunogenicity, which refers to the generation of antigen-specific antibodies [humoral immunity] and antigen-specific T-cells [cellular immunity], with immunostimulation, which refers to cytokine secretion [cytokine storm if too high levels of cytokines are produced), complement activation, TLR stimulation. Many published papers are confusing and misleading because they use the term "immunogenicity" when describing cytokine responses, i.e. "immunostimulation."

We thank the reviewer for this side note. We understand the importance of avoiding confusion between immunogenicity and immunostimulation. We have checked our entire manuscript to make sure we are not using immunostimulation and that we are correctly using the term immunogenicity.

Response to Reviewer #2:

The authors did a nice job addressing my concerns and some of the other reviewers' concerns.

We thank the reviewer for their positive assessment of our revised manuscript.

I don't see the CpG conjugation efficiency validation. Based on our study, it's not always 100% conjugated.

We would like to apologize for missing the CpG conjugation efficiency characterization. We agree with the reviewer that the coverage is not always 100%. To report the exact coverage, we have performed new experiments with a fluorescently labelled version of the CpG, in order to determine the conjugation efficiency of CpG on the DNA-NPs displaying 10 overhangs for the conjugation of CpGs. The new results have been included in Supplementary Figure 15 (Page S23 of the supplementary information document) and in a new paragraph in the Results and discussion section (page 10 of the main manuscript, see first text box below) and added the specific method used in the Methods section (Page 23 of the main manuscript, see second text box below).

Quantification of the coverage of CpG on DNA-NP. The quantification of CpGs on DNA-NPs was determined using fluorescence measurements with a fluorescein-labelled version of the CpG ODN. Using a standard curve made with the free fluorescein-CpG ODN, we estimated the CpG ODN hybridization yield to be about 80% (Supplementary Figure 15).

Assessment of CpG coverage yield on NP via measuring fluorescein emission. Fluorescence-intensity based quantitate measurement was applied to ascertain the approximate number of CpG hybridization over DNA-NP. The standard curve was obtained based on the emission spectra of fluorescein in varying concentration of fluorescein-labelled CpGs (0 to 10 μ M). The excitation wavelength was set at 475 nm and emission wavelength was set between 500 to 700 nm with a maximum emission peak at 520 nm. According to the fluorescence intensity of a reference fluor-CpG ODN concentration, the number of modifications over DNA-NP by CpG ODN was estimated for DNA-NP samples modified with the fluorescently labelled CpGs as described earlier in the method section.

I also have one question for figure 6c. What are the comparing groups there? Shouldn't it be the different vaccine formulas you are comparing? I don't see the need to compare the different Igs.

We apologize for the confusion with Fig 6c. Here we are only comparing the most efficient formulation, as determine in the viral challenge assay with an injection of PBS that we called “placebo” in the previous version of our manuscript. To avoid any confusion we replaced the term Placebo with PBS in the Figure 6 and edited the corresponding caption.

We are not comparing the different Igs level but rather we are just providing the values for a single group of mice injected with the most efficient formulation of our vaccine as determined by the challenge assay.

Another minor: in the method section for control groups, could you also add the amount of CpG and nanoparticles beside the amount of antigen?

Thanks for the suggestion and sorry for missing these details. We have updated the Methods section accordingly (Page 20).